# Mechanisms governing the pioneering and redistribution capabilities of the non-classical pioneer PU.1

Julia Minderjahn[1], Andreas Schmidt[2], Andreas Fuchs [3], Rudolf Schill[4], Johanna Raithel[1], Magda Babina [5], Christian Schmidl [6], Claudia Gebhard[1,6], Sandra Schmidhofer[1,7], Karina Mendes[1], Anna Ratermann[1,8], Dagmar Glatz[1,9], Margit Nützel[1], Matthias Edinger[1,6], Petra Hoffman[1,6], Rainer Spang[4], Gernot Längst [3], Axel Imhof [2] & Michael Rehli [1,6]*

Establishing gene regulatory networks during differentiation or reprogramming requires master or pioneer transcription factors (TFs) such as PU.1, a prototype master TF of hematopoietic lineage differentiation. To systematically determine molecular features that control its activity, here we analyze DNA-binding in vitro and genome-wide in vivo across different cell types with native or ectopic PU.1 expression. Although PU.1, in contrast to classical pioneer factors, is unable to access nucleosomal target sites in vitro, ectopic induction of PU.1 leads to the extensive remodeling of chromatin and redistribution of partner TFs. De novo chromatin access, stable binding, and redistribution of partner TFs both require PU.1's N-terminal acidic activation domain and its ability to recruit SWI/SNF remodeling complexes, suggesting that the latter may collect and distribute co-associated TFs in conjunction with the non-classical pioneer TF PU.1.

[1] Department of Internal Medicine III, University Hospital Regensburg, 93053 Regensburg, Germany. [2] Biomedical Center, Protein Analysis Unit, Faculty of Medicine, Ludwig-Maximilians-Universität München, Großhaderner Strasse 9, 82152 Planegg-Martinsried, Germany. [3] Biochemistry Centre Regensburg (BCR), University of Regensburg, 93053 Regensburg, Germany. [4] Statistical Bioinformatics Department, Institute of Functional Genomics, University of Regensburg, 93053 Regensburg, Germany. [5] Department of Dermatology and Allergy, Charité Universitätsmedizin Berlin, Berlin, Germany. [6] Regensburg Center for Interventional Immunology (RCI), University Regensburg and University Medical Center Regensburg, 93053 Regensburg, Germany. [7] Present address: AstraZeneca, Tinsdaler Weg 183, 22880 Wedel, Germany. [8] Present address: Rentschler Biopharma SE, 88471 Laupheim, Germany. [9] Present address: Chromatin Structure and Cellular Senescence Research Unit, Maisonneuve-Rosemont Hospital Research Centre, Montréal, QC, Canada H1T 2M4. *email: michael.rehli@ukr.de

Cellular differentiation requires so-called master or pioneering transcription factors (TFs) to establish access to regulatory elements embedded in chromatin[1]. Similar to the vast majority of TFs, they usually recognize specific DNA motifs ranging from 6 to 12 base pairs (bp) in length, implying the existence of roughly a million potential binding sites throughout the genome[2]. How the actually bound sites, which usually range in the thousands, are selected from the vast array of putative binding sites is largely unknown.

The ETS family TF PU.1 (encoded by *SPI1*) is a well-studied master regulator of the hematopoietic compartment and is required for the generation of common lymphoid and granulocyte-macrophage (MAC) progenitor cells, as well as later stages of monocyte (MO)/MAC and B-cell development[3]. It controls the expression of hundreds of genes that include growth factor receptors, adhesion molecules, TFs, and signaling components[4], and is able to initiate myeloid gene expression programs in heterologous cell types including T cells and fibroblasts[5]. The ability of PU.1 to shape chromatin landscapes and re-program cells and its role in regulating cell type-specific gene expression make it a prototypic pioneer factor[6]. Yet, how this factor interacts with chromatin to access its binding sites de novo to date has not been elucidated.

The first genome-wide analyses of PU.1 occupancy observed cell type-specific binding-site selection in murine MACs and B cells[7,8], as well as in human MOs and MO-derived MAC[9]. Cell type-specific binding sites depended on the co-occurrence of sequence motifs for other cell type-specific TFs, including members of the C/EBP and AP-1 family TFs in human MO or murine MAC, EGR2 in human MO-derived MAC, or E2A, EBF, and OCT2 in B cells[8,9]. The exact mechanisms behind this cooperativity between PU.1 and other TFs are not well understood and may include direct protein–protein interactions, interactions between TFs that are facilitated by DNA, DNA-mediated interactions in the absence of TF interactions, and indirect cooperativity involving competition between nucleosomes and TFs[10]. The requirement for TF cooperativity is inversely correlated with motif affinity: high-affinity motifs are frequently bound by PU.1 alone, whereas low-affinity motifs are only bound when other factors co-bind nearby[11]. Known physical interaction partners of PU.1 include general TFs such TFIID and TBP (TATA-box binding protein), cell type-specific TFs such as interferon regulatory factor 4 and 8 (IRF4 and IRF8), the proto-oncogene c-Jun (JUN, a component of the AP-1 TF), and early hematopoietic TFs such as GATA-binding protein 2 (GATA2) and runt-related TF1 (RUNX1)[12–17]. In early T cells, PU.1–RUNX1 interactions lead to a redistribution of RUNX1 binding, highlighting the importance of TF interactions as well as TF protein levels in binding-site selection[18]. These interactions, however, do not explain how PU.1 exerts its presumed pioneering role or how it selects its binding sites in chromatin in the first place.

Here we systematically analyze the ability of PU.1 to access its binding sites in vitro and in vivo. By profiling PU.1 binding across a large atlas of hematopoietic cell types, we show that PU.1 only occupies a fraction of its potential binding sites, and that cell type-specific binding is not exclusively explained by TF co-association. In vitro studies further show that PU.1 binding to DNA is subject to both epigenetic and chromatin constraints. It is unable to bind CpG-methylated or nucleosome-bound DNA, suggesting that PU.1 may not act as a classical pioneer factor, which are defined by their ability to recognize their binding sites in nucleosomal DNA. Despite these constraints, introduction of PU.1 into heterologous model cell lines lacking endogenous PU.1 expression leads to extensive de novo remodeling of chromatin at PU.1-binding sites and rapid initiation of a myeloid gene

expression program. Functional analysis of several mutant PU.1 variants indicates that efficient binding of PU.1 to de novo-remodeled sites depends on the N-terminal acidic activation domain (AAD), suggesting that the latter is strictly required for accessing binding sites de novo. Further analyses including in vivo proximity-dependent biotinylation (BioID) and co-immunoprecipitation (CoIP) show that the N-terminal acidic domain mediates the interactions of PU.1 with the SWI/SNF family of chromatin remodeling complexes (BAF complex). Hence, the ability of PU.1 to shape regulatory landscapes and to act as a non-classic pioneer factor requires the AAD and its interaction with SWI/SNF. The redistribution of partner TFs by PU.1 also requires the SWI/SNF-interacting acidic domain, suggesting that the remodeler complex may act as part of a hub to collect and distribute co-associated TFs in a PU.1-dependent manner.

## Results

**PU.1 binding across multiple cell types in vivo and in vitro.** To better understand what distinguishes PU.1-bound sequences from unbound sequences, we first determined its DNA-binding profiles across a large array of different cell types. PU.1 DNA-binding maps have already been generated in a number of studies, but comparative analyses were generally restricted to few cell types. For a more comprehensive view of PU.1-binding patterns, we collected publicly available occupancy data and generated several additional PU.1 binding maps in various lymphoid and myeloid cell lines, primary human cells and several MO-derived cell types (Fig. 1a). Summaries of published and generated chromatin immunoprecipitation sequencing (ChIP-seq) data are provided in Supplementary Table 1 and Data File 1. With reference to a previously defined PU.1 consensus sequence[11], PU.1 binding (defined by standard or stringent peak calling criteria) was only detected at a fraction (<20%) of possible binding sites (Fig. 1b). Bound sites were generally characterized by higher motif scores (Fig. 1c). Cell type-restricted binding events (comprising 27% of stringent peaks) were enriched for co-associated sequence motifs, suggesting that combinatorial TF interactions support binding at low-affinity sites (Supplementary Fig. 1d). Notably, high-affinity sites (high motif log odds scores) are frequently bound across the large majority of cell types (Supplementary Fig. 1e) and are almost absent from the group of motifs showing no evidence of ChIP-seq signals (Fig. 1c). Nevertheless, the observed dependency on DNA sequence motifs (both for PU.1 and partner TFs) was not mandatory, suggesting that PU.1 binding is controlled on additional levels.

Obvious candidate mechanisms include the possible restriction of PU.1 binding to DNA via the covalent modification of DNA (e.g., DNA methylation) or via the competition with nucleosomes. To test this in vitro, we performed binding assays with recombinant PU.1 on either (hydroxy-)methylated or nucleosome-bound double-stranded DNA (dsDNA). DNA (hydroxy-)methylation was found to interfere with PU.1 binding when present close to the GGAA-core sequence, but not further downstream, and only when occurring on the sense strand (Fig. 1d). Of all sequences covered by the PU.1 consensus, only a small fraction of about 20 K (10% of which are located in promoters) contain a proximal CpG that might affect binding. Nucleosomes also presented a barrier to PU.1 binding. Although sites located around to the nucleosome dyad axis were not bound at all, sites proximal to the nucleosome entry site were at least partially accessible to PU.1 (Fig. 1e). This is in line with published data for the highly homologous ETS domain of SPIB, which showed a clear binding preference for linker and nucleosome entry sites in the NCAP-SELEX assay[19]. These in vitro binding

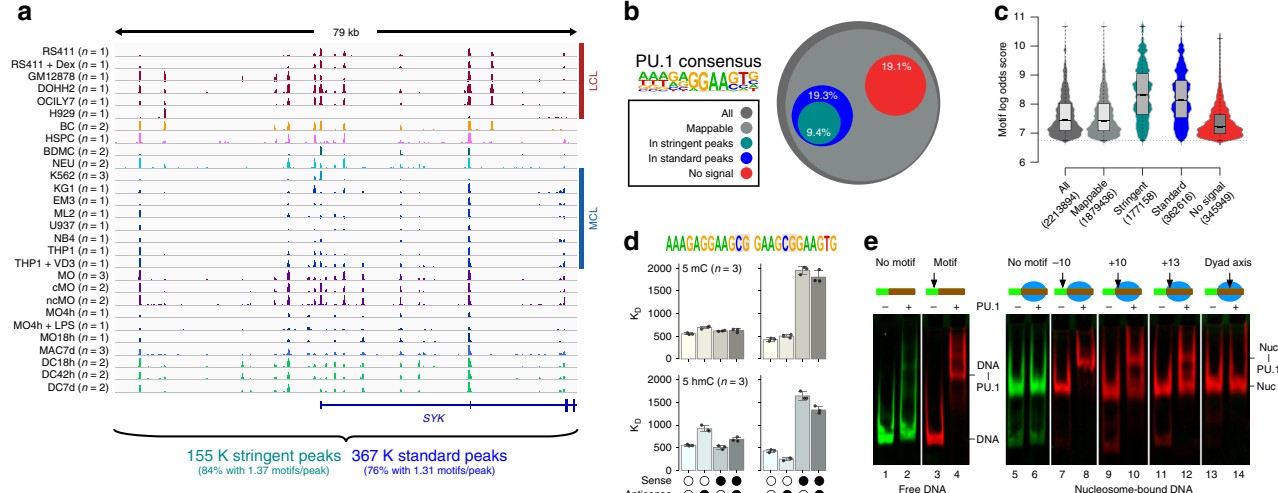

**Fig. 1 PU.1 occupancy in vivo and binding constraints in vitro. a** Comparison of PU.1 ChIP-seq data across various human lymphoid and myeloid cell lines (LCL and MCL, respectively) and primary cells (BC B cells, BDMC breast skin-derived mast cells, cMO classical MO, DC dendritic cells, HSPC hematopoietic stem and progenitor cells, MAC, macrophages, MO monocytes, ncMO non-classical MO, NEU neutrophils) at an exemplary locus. For replicated data sets (as indicated), averaged coverage tracks are shown. Total motif occurrences in standard and stringent peaks are summarized below the tracks. **b** Fraction of PU.1 motifs residing in either standard or stringent PU.1 peaks ($3.61 \times 10^5$ and $1.77 \times 10^5$, respectively) compared with all motif occurrences ($2.21 \times 10^6$) across the genome, motifs filtered for mappability ($1.88 \times 10^6$), as well as motifs that showed no evidence of binding (<3 per $10^7$ reads within 200 bp motif-centered window) across all samples (no signal, $3.48 \times 10^5$). **c** Distribution of motif scores across total occurrences, mappability-filtered motifs, motifs in peaks, and no-signal motifs. **d** Microscale thermophoresis-derived dissociation constants ($K_D$ values, bars represent the mean of $n = 3$ experiments ± SD, individual data points are shown as black dots) for the interaction between recombinant full-length PU.1 and the indicated double-stranded oligonucleotides representing PU.1 motifs with a high motif log odds score (left panels, 9.4; right panel 8.8). Open circles indicate unmethylated cytosines. Black circles represent methylated or hydroxymethylated cytosines (top panels, 5mC or bottom panels, 5hmC, respectively). **e** Gelshift assays demonstrating position-dependent binding of PU.1 to nucleosome-associated DNA. The relative position of the PU.1-binding site (arrow) to the nucleosome boundary is indicated. Nucleosomes are positioned by the 601 nucleosome-positioning sequence marked in red. The positions of free DNA (DNA) and nucleosomes (nuc), as well as of DNA-PU.1 and nuc-PU.1 complexes are indicated. **a–d** Source data are provided as a Source Data file.

studies suggested that PU.1 alone is indeed restricted by both DNA methylation and chromatin. Given the well-documented pioneering role of this TF and its ability to access a large fraction of high-affinity sites in vivo, PU.1 must be able to overcome epigenetic and chromatin constraints at least to some extent.

**Immediate consequences of de novo PU.1 expression.** PU.1 occupancy maps across multiple cell types were useful to explore cell type-specific binding preferences, but less informative regarding rules of de novo binding-site selection. To better understand how PU.1 selects binding sites in vivo and which auxiliary factors may help to overcome epigenetic and chromatin constraints, we established a system where PU.1 expression is rapidly induced in a PU.1-negative acute lymphoblastic leukemia cell line (CTV-1) using mRNA transfection (Fig. 2a, b). CTV-1 cells neither express endogenous PU.1 nor its related class III ETS family members SPIB and SPIC (Fig. 2c), suggesting that recognition sites of class III ETS factors may be "untouched" in these cells. IRF4 and IRF8, which may form heterodimers with PU.1, were also not detected in CTV-1 cells (Fig. 2c). At its peak, PU.1 protein expression levels in CTV-1 after mRNA transfection exceeded those of natively high expressing cell types (such as MO-derived dendritic cells (DCs) or THP-1 cell line; Supplementary Fig. 2a).

In this model system, we observed de novo chromatin remodeling and transcription (Fig. 2d, e) upon PU.1 expression. Induced genes (measured 24 h after PU.1 mRNA transfection) were associated with Gene Ontology (GO) terms such as myeloid activation and differentiation (Fig. 2f), and expression levels of

these genes correlated with PU.1 expression across hematopoietic cells types (Fig. 2g).

After establishing that transient PU.1 mRNA transfection in CTV-1 cells affects relevant target genes, we examined the effects of PU.1 expression on chromatin. Relative to control-transfected cells (PU.1mut), we detected 45 K PU.1-binding sites (using stringent peak calling criteria) by ChIP-seq. Of those, 80% or 94% overlapped with stringent or standard peaks sets derived from all natively PU.1-expressing cell types (in Fig. 1a), respectively. The number of detected peaks was comparable to other highly PU.1-expressing cell types (e.g., 51 K stringent peaks in DC, 35 K stringent peaks in MO-derived MACs, or 63 K stringent peaks in phorbol 12-myristate 13-acetate (PMA) and Vitamin D3-treated THP-1 cells; peak counts of all samples are listed in Supplementary Table 1 and Data File 1). As shown in Fig. 3, clustering of PU.1 peaks based on corresponding ATAC-seq data (a measure of chromatin accessibility) separated PU.1 peaks into groups of peaks with different degrees of accessibility before and after PU.1 induction and revealed extensive chromatin remodeling upon PU.1 expression at PU.1 peak clusters 1–8, comprising the large majority of PU.1-binding sites. PU.1 peak clusters with highly remodeled sites (e.g., PU.1 peak clusters 6–8) were significantly co-associated with the induced expression of neighboring genes, either after a single PU.1 induction (short), or seven cycles of PU.1 mRNA electroporation over 7 days (long).

To further characterize the properties of PU.1 peaks in clusters defined by chromatin accessibility (Fig. 3), we first analyzed their motif composition. We found that de novo-remodeled sites generally associate with higher-affinity motifs compared with PU.1-binding sites that were located in accessible chromatin prior

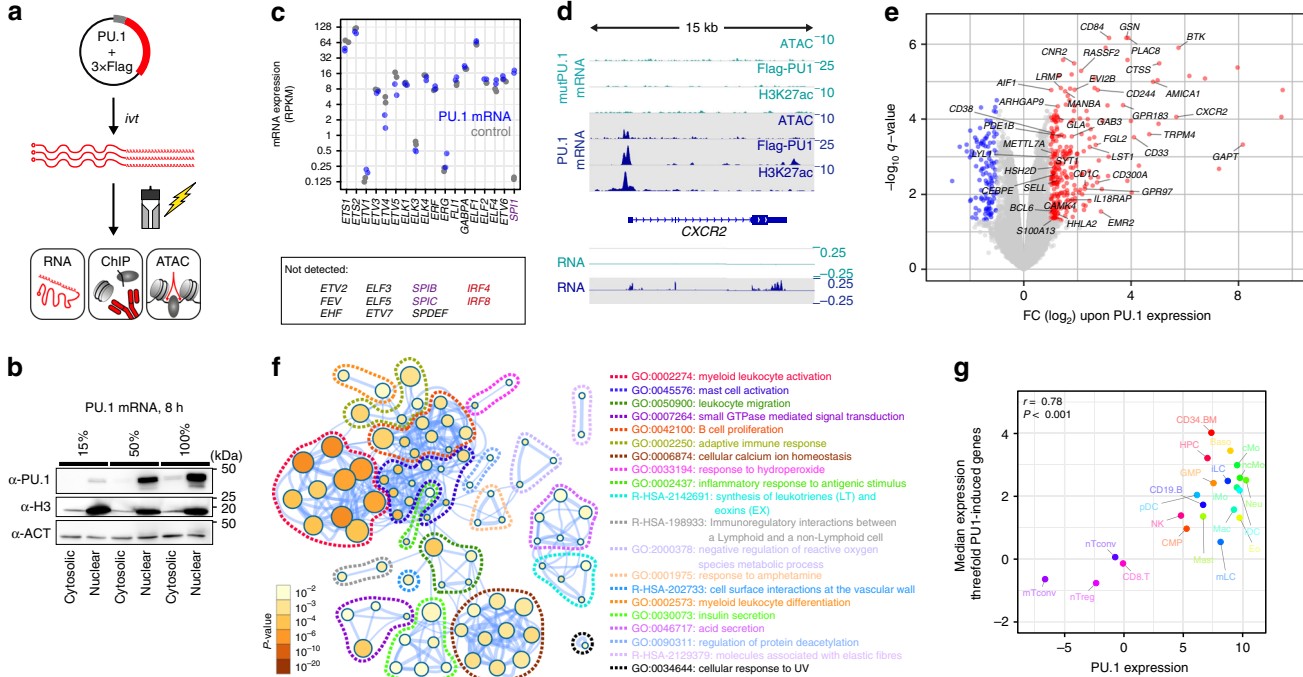

**Fig. 2 Transient PU.1 expression induces a myeloid expression program. a** Schematic of the experimental setup. **b** Immunoblotting confirming the translation and nuclear location of full-length PU.1 in transfected cells using increasing amounts of IVT mRNA. **c** Relative mRNA expression levels of ETS family members in PU.1-expressing or control cells measured by RNA-seq (n = 2 each). **d** Example tracks for a PU.1-activated gene. Coverage represents the mean of n = 3 or n = 2 experiments for ATAC/ChIP-seq or RNA-seq data, respectively. **e** Volcano plot of significantly PU.1 up- or downregulated genes (red or blue, respectively). Genes with known function in myeloid or B-cell biology are labeled. **f** Metascape-derived network of Gene Ontology (GO) terms associated with PU.1-induced genes in CTV-1 cells. The significance of enrichment of a particular term is depicted with the log10 of the p-value and indicated by node coloring. **g** Comparison of median expression levels of PU.1-induced genes (threefold) with PU.1 expression levels across various human blood cell types data based on CAGE (Cap Analysis of Gene Expression) expression data from the FANTOM Consortium[35]. Related cell types in the hematopoietic tree share similar colors. The Pearson's correlation as well as the corresponding p-value are indicated. **b**, **c**, **e**–**g** Source data are provided as a Source Data file.

to PU.1 induction (Fig. 4a and Supplementary Fig. 2b). The increasing chromatin accessibility through PU.1 peak clusters 1–14 was accompanied by a linear drop in average PU.1 motif log odds scores and the increasing presence of nearby motifs for co-associated TFs such as RUNX or GATA (Fig. 4b, c and Supplementary Fig. 2c), which likely support the binding of PU.1 at less-affine sites. In line with previous observations[11], a high degree of accessibility also correlated with evolutionary conservation (Fig. 4d) and genomic annotation as promoters (Fig. 4e), whereas de novo-remodeled sites often showed little conservation across species and were mostly inter-/intragenic (Fig. 4d, e). Although the above suggested an important role of sequence-related features in the PU.1-binding-site selection, modeling revealed that motif scores, conservation, and nearby presence of co-associated motifs poorly discriminated PU.1-bound and -unbound sites (Fig. 4f). Models including chromatin accessibility in control cells showed a marginally improved predictive power (Fig. 4f), suggesting that additional features (such as chromatin structure or epigenetic modifications) must have a significant impact on PU.1-binding-site selection. The best predictor included chromatin accessibility data after PU.1 induction (Fig. 4f), which is in line with the ability of PU.1 to increase accessibility at a large majority of binding sites.

Further analysis of chromatin accessibility data at single-nucleotide resolution revealed common footprints across PU.1 motif-centered peaks in PU.1 peak clusters (examples are shown in Fig. 4g). Notably, in PU.1 peak clusters with pre-accessible chromatin (peak clusters 12–14), the footprints across PU.1 motifs pre-existed. The lack of change in accessibility, despite the

de novo binding of PU.1 suggests replacement or competition with other (likely ETS family) TFs at these elements (Fig. 4g, right histogram) rather than assisted loading as observed for GR[20].

PU.1 induction also caused the rapid disappearance of ~3 K accessible sites (Fig. 3, bottom panel), which were highly enriched for consensus TF motifs (like RUNX, ETS, or GATA) that were also identified in PU.1-induced accessible sites (Fig. 5a). The loss of corresponding footprints at former and their gain at latter sites (Fig. 5b–d) suggests the PU.1-induced redistribution of these partner TFs, as recently observed for RUNX1 and SATB1 in early T cells[18].

To further confirm the marked impact of PU.1 induction on chromatin landscapes, we analyzed two additional heterologous cell lines (T-cell acute lymphoblastic leukemia-1 (TALL1) and the hepatocellular carcinoma cell line HepG2). As observed in CTV-1 cells, PU.1 induction in either cell line caused the rapid reorganization of chromatin landscapes, which showed similar associations with either high-affinity motifs or the presence of cell type-specific co-associated motifs, such as RUNX, IRF, and E2A motifs in TALL1 cells, and FOXA1, HNF1, and HNF4 motifs in HepG2 cells (Supplementary Fig. 3a–e). The predictive power of individual features shown in Supplementary Fig. 3f was also similar to CTV-1 cells, indicating similar constraints for PU.1 binding. Using published whole genome DNA methylation data for HepG2 cells, we could additionally confirm the almost exclusive binding of PU.1 to unmethylated GGAA-core sequences (Supplementary Fig. 3g), as predicted from the in vitro binding studies (shown in Fig. 1d). Redistribution of cell type-specific TFs was also clearly evident in both cell types (Supplementary

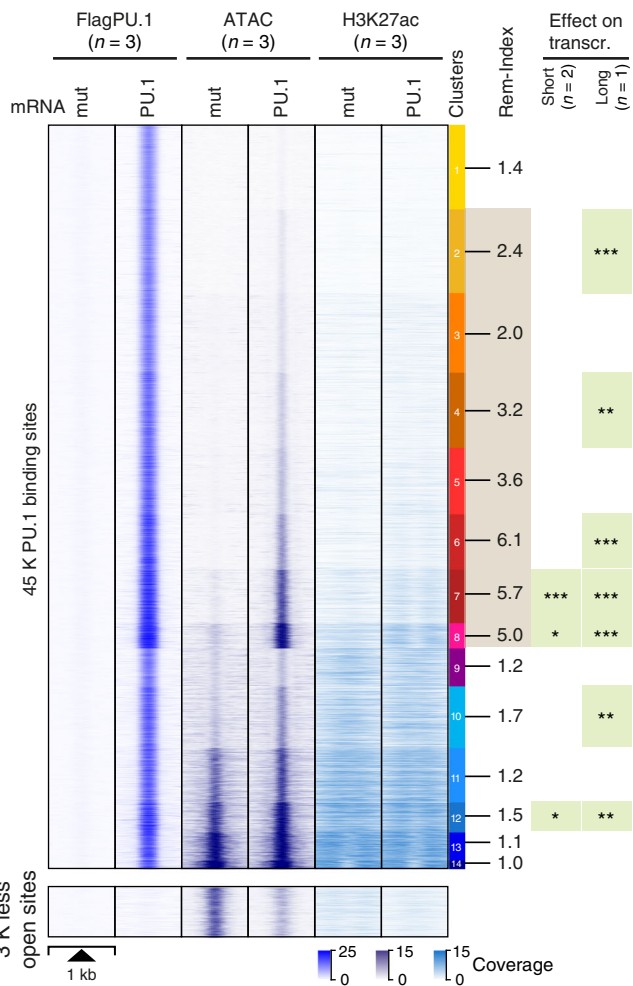

**Fig. 3 PU.1-induced changes in chromatin accessibility.** The distribution of average PU.1 ChIP-seq, ATAC-seq, and H3K27ac ChIP-seq signals of PU.1 vs. PU.1mut mRNA-transfected cells 8 h after electroporation are plotted across 1 kb windows and 45 K PU.1-binding sites (top panel) or 3 K regions that lost accessibility after PU.1 induction (bottom panel) in CTV-1 cells. PU.1 peaks are ordered according to K-means clustering of peak-centered ATAC-seq signals. PU.1 peak clusters are indicated by the color bar on the right, along with the average remodeling index (Rem-Index) of each cluster. The asterisks on the right indicates the significant induction of mRNA expression across genes associated with PU.1 peaks in the indicated peak cluster (***P < 0.001; **P < 0.01; *P < 0.05; paired Wilcoxon's test) in PU.1-expressing CTV-1 cells. The coloring of PU.1 peak clusters is kept consistent in all following analyses based on the clustering. Source data are provided as a Source Data file.

Fig. 3h–k), suggesting that PU.1 operates similarly in different cell types.

**Interaction among ETS family factors**. As part of the motif enrichment analysis (Fig. 4c), we noted that de novo-remodeled PU.1 peaks frequently contained two (or more) PU.1 motifs. Across peaks, these homotypic motif pairs were significantly enriched (compared with non-bound motifs) in a range of 12–50 bp (Supplementary Fig. 4a), which resembles early findings of clustered PU.1-binding sites in many myeloid-specific promoters[21]. Notably, homotypic motif pairs were preferentially found in the de novo-remodeled fraction of peaks (PU.1 peak clusters 1–8). Their appearance correlated with the degree of remodeling across PU.1 peak clusters (Supplementary Fig. 4b–f), suggesting that PU.1 motif pairing could assist binding-site

selection. However, the motifs in pairs also showed significant overlaps with consensus sequences for other ETS class family members (Supplementary Fig. 4g), suggesting that de novo-remodeled "homotypic" motif pairs are either bound by PU.1 alone or in combination with another ETS family member expressed in CTV-1 cells.

As homotypic or heterotypic interactions of PU.1 may influence binding-site selection, we asked whether and how the distribution of other ETS factors would be affected by PU.1 expression and to which extent binding sites of PU.1 overlapped with other ETS factors. We generated occupancy maps for two prominent, representative ETS factors (FLI1, ETS1) in CTV-1 cells before and after PU.1 expression (an exemplary locus is shown in Fig. 6a). Both factors share an almost identical ETS class 1a motif, which was clearly different from the PU.1 consensus site (Fig. 6b, c). Correspondingly, the largest fraction of FLI1 and ETS1 target sites overlapped (Fig. 6d), suggesting that they compete for the majority of genomic binding sites. The small fraction of ETS1-specific peaks was primarily associated with two co-associated motifs (ZNF143 and a composite ETS:RUNX motif, see Supplementary Fig. 5a, b), indicating that corresponding TFs may favor ETS1 at these sites.

The induction of PU.1 had a major impact on the genomic distribution of FLI1 and ETS1 in CTV-1 cells (Fig. 6e, f and Supplementary Fig. 5c). As already indicated by the footprints across PU.1 motifs observed at pre-accessible PU.1-binding sites (as shown for PU.1 peak cluster 13 in Fig. 4g), PU.1 joined the competition of ETS factors at a large fraction of pre-existing ETS-binding sites (across clusters 9–14). Correspondingly, the ChIP-seq coverage of ETS1 and FLI1 at pre-accessible PU.1-binding sites was reduced after PU.1 induction (Supplementary Fig. 5d).

Likewise, both ETS factors joined PU.1 at a major subset of de novo-remodeled fraction of peaks (Fig. 6e, f and Supplementary Fig. 5c, f) across PU.1 peak clusters 1–8. Motif scores of ETS factors and PU.1 at their binding sites showed an inverse correlation across de novo-remodeled PU.1 peak clusters 1–8 (Fig. 4a and 6g, and Supplementary Fig. 5e). The predicted recognition motif resembled the ETS motif at PU.1-binding sites co-bound by ETS factors (both at single and paired motif sites), whereas sites without evidence of ETS binding resembled the PU.1 consensus motif (Supplementary Fig. 5g). This suggests that the ETS factor distribution is driven at least in part by motif affinities of individual factors. At sites bound by PU.1 alone, chromatin accessibility changes were limited, regardless of the presence of single or paired sites (Supplementary Fig. 5h), suggesting that binding at these motif pairs is likely restricted to a single position. At present, we cannot say whether the recruitment of ETS factors (or other partner factors) to de novo-remodeled sites actively contributes to the process of remodeling or whether it stabilizes the accessible space between two nucleosomes created in the course of PU.1 binding. Nevertheless, it is clear that at these sites, PU.1 is required to allow for ETS factor binding, which is not observed in the absence of PU.1.

In line with the redistribution of other partner TFs (as shown in Fig. 5a–d), the binding of ETS1 and FLI1 was also reduced at the disappearing ~3 K sites that were accessible prior to PU.1 induction (Fig. 6e, bottom panel), further corroborating the ability of PU.1 to redistribute other TFs.

**PU.1 domains and interactors required for de novo binding**. Next, we sought to characterize the mechanism underlying PU.1's ability to change chromatin accessibility and redistribute other TFs across the genome. Hypothesizing that specific interactions of PU.1 with other proteins are involved in these changes, we tested the effects of deleting different known protein–protein interaction

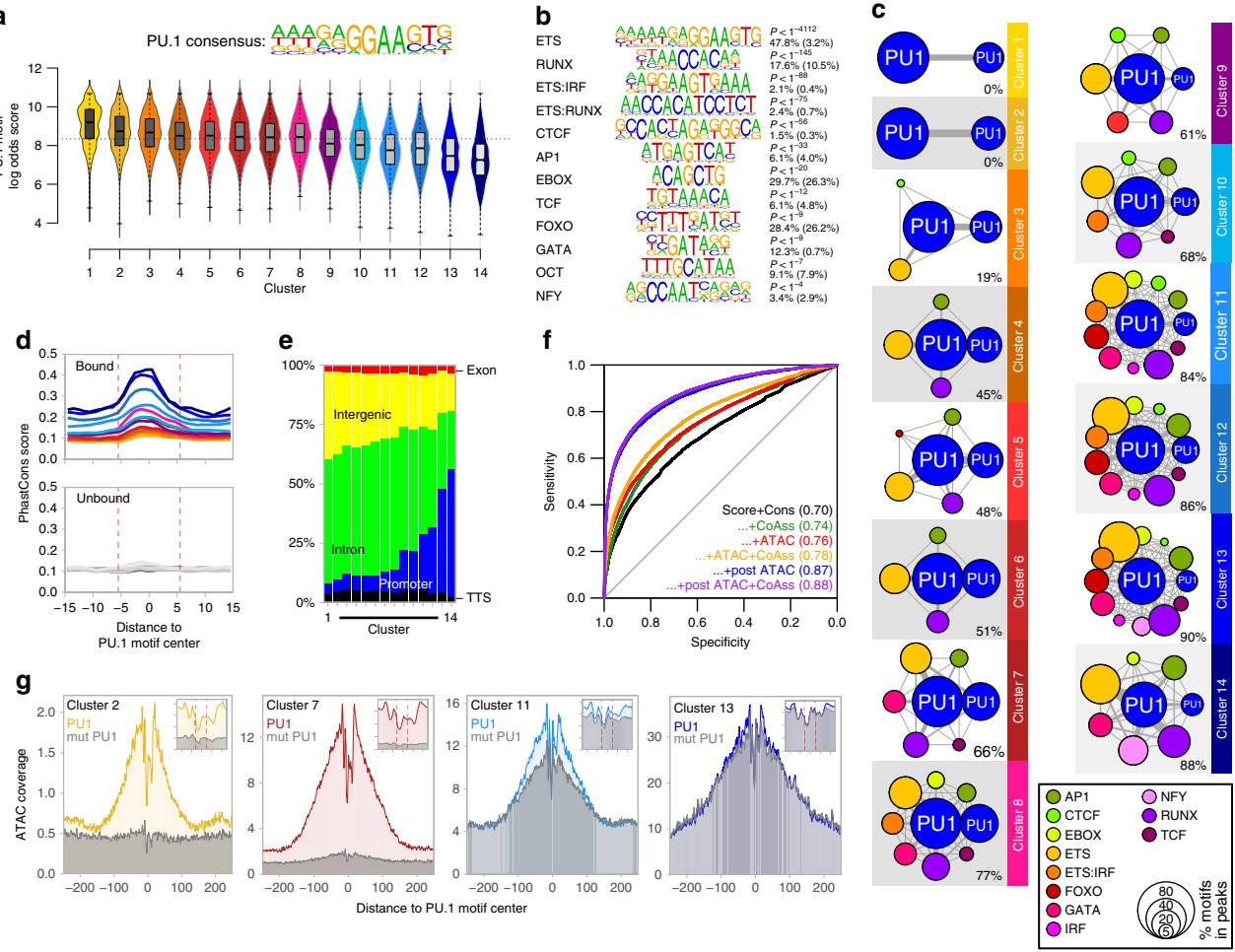

**Fig. 4 Genomic features of PU.1-induced accessible sites. a** The PU.1 motif log odds score distribution of the consensus MAC PU.1 motif, generated earlier, is shown for each cluster. The median of the specific distribution across all clusters is depicted inside the bean with a conventional boxplot. **b** De novo-derived co-associated motifs enriched across ATAC-seq-derived PU.1 peak clusters. The best motifs corresponding to known factor families derived from individual clusters are shown. Significance of motif enrichment (hypergeometric test) and the fraction of motifs in peaks (background values are given in parenthesis) correspond to their distribution across all PU.1 peak regions. **c** Motif co-enrichment networks for individual clusters are shown. The size of each node represents the motif enrichment (fraction of peaks) and co-associated TF motifs (PU.1 masked) are indicated by coloring. The second PU.1 node corresponds to the fraction of peaks containing at least two PU.1 motifs. Edge thickness indicates the frequency of motif co-association within the PU.1 peak. The fraction of PU.1 peaks overlapping with co-associated (PU.1 masked) TF motifs is given below each network. **d** Evolutionary conservation of the PU.1 motif across the K-means clusters, color coded as in Fig. 3 as illustrated by the PhastCons score. Corresponding sequence-matched random control sets of non-bound motifs are shown in the bottom histogram (in dark gray to light gray). **e** Genome ontology analysis across the K-means clusters is shown in a stacked bar chart. The association with the individual regions is given as the fraction of peaks in each cluster. **f** Predictability of PU.1 binding from motif score, conservation score, nearby presence of co-associated motifs, and chromatin accessibility before and after PU.1 induction. Shown are ROC curves of logistic models with different sets of predictors (AUC in parentheses), trained and evaluated on separate subsets of the data. **g** Representative ATAC-seq footprints across motif-centered, cluster-associated peaks (as indicated by coloring). Corresponding footprints of control cells (mutPU.1) are shown in gray. Smaller histograms in the upper right corner zoom into the central part of the main graph. The position of the PU.1 motif is indicated by two vertical dashed lines. **a–g** Source data are provided as a Source Data file.

domains of PU.1 on its ability to access chromatin de novo. To this end, we generated PU.1 expression constructs devoid of each one or all of the acidic (A), glutamine-rich (Q), and PEST (P) domains, and tested them in our model compared with wild-type PU.1 (Fig. 7a). All mutant proteins were expressed and translocated to the nucleus as expected (Fig. 7b). Detected peak sets of PU.1 mutants generally represented subsets of the wild-type and in three of four cases, binding profiles differed significantly from wild-type PU.1 (Fig. 7c, d). Most pronounced changes in peak coverage were observed for the isolated ETS domain (ΔAQP, 77% of wild-type PU.1 peaks showed significantly reduced signals) and the mutant lacking the acidic domain (ΔA, 72% of wild-type PU.1 peaks showed significantly reduced signals), followed by the less-

affected mutant lacking the glutamine-rich domain (ΔQ, 52% of wild-type PU.1 peaks showed significantly reduced signals), which partially resembled a lower dose of PU.1 (15% PU.1, 29% of wild-type PU.1 peaks showed significantly reduced signals). The ΔP mutant did not reveal significantly altered binding patterns. Interestingly, binding profiles of ΔA and ΔQ mutants were particularly different at highly remodeled clusters (Fig. 7e, clusters 7/8), suggesting that the ΔA mutant may specifically lack remodeling capacity.

To identify interaction partners of PU.1 that may explain the observed effect of PU.1 mutant proteins, we adapted the BioID approach[22] to biotinylate and identify candidate interacting proteins in the proximity of PU.1 (Fig. 8a). We initially validated

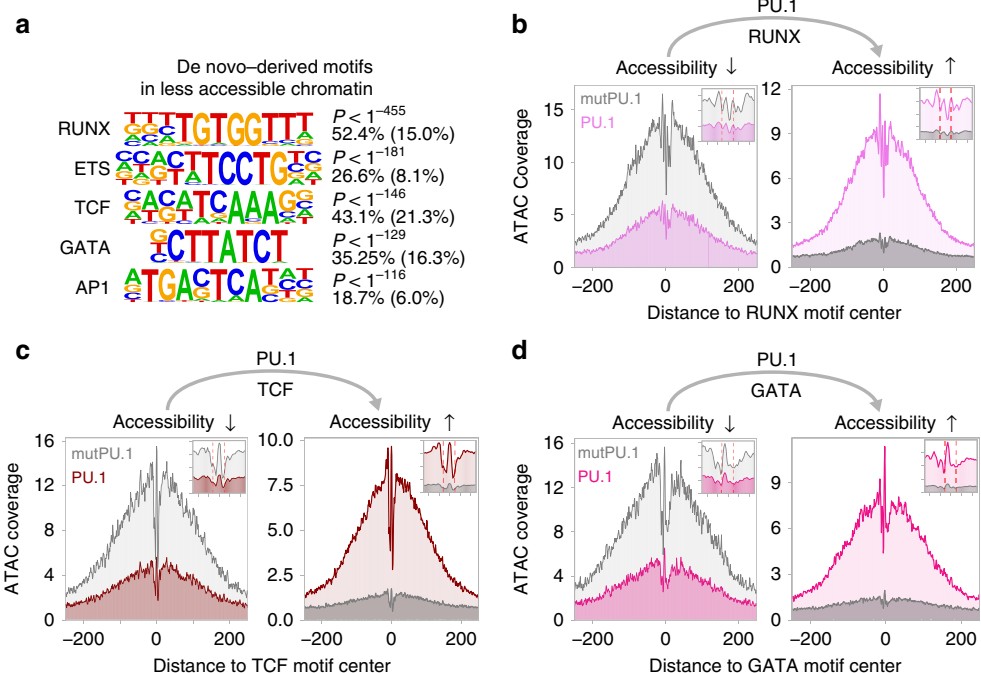

**Fig. 5 Motif signatures at sites loosing accessibility. a** De novo-derived motifs enriched across regions that lost accessibility upon PU.1 induction in CTV-1 cells. The best motifs corresponding to known factor families derived from individual clusters are shown. The significance of motif enrichment (hypergeometric test) and the fraction of motifs in peaks (background values are in parenthesis) are given for the top five motifs corresponding to known motif families. **b–d** Representative ATAC-seq footprints across enriched motif-centered peaks that either lost or gained accessibility upon PU.1 induction in CTV-1 cells. Footprints of control cells (mutPU.1) are in grey, footprints of PU.1 mRNA-transfected cells are colored. Smaller histograms in the upper right corner zoom into the central part of the main graph. The position of the PU.1 motif is indicated by two vertical dashed lines. **a–d** Source data are provided as a Source Data file.

that the BirA fusion constructs were expressed, functional, and recruited to the same locations compared with wild-type constructs (Supplementary Fig. 6a–c). BioID experiments in CTV-1 cells transfected with wild-type PU.1-BirA and BirA carrying a nuclear localization sequence (NLS) as control (NLS-BirA) followed by mass spectrometry of biotin-ligated proteins revealed significant enrichment of proteins associated with the SWI/SNF complex (Fig. 8b, c) in the neighborhod of PU.1. We obtained similar results with constitutively PU.1-expressing THP-1 and K-562 cells, confirming that this was not confined to CTV-1 cells (Supplementary Fig. 6d). Performing BioID analysis with the ΔA and ΔQ deletion mutants, we found that the ΔA mutant, but not the ΔQ mutant, specifically lost proximity to the SWI/SNF remodeling complex relative to full-length PU.1 (Fig. 8d). The specific interaction between PU.1 and SWI/SNF was also observed in CoIP experiments. Focusing on the central ATP-dependent chromatin remodeler SMARCA4 (BRG1), we observed its specific interaction with PU.1 independent of the presence or absence of DNA (Supplementary Fig. 6e), and both in FLAG-PU.1 and BRG1 CoIPs (Fig. 8e and Supplementary Fig. 6f). We also detected two additional SWI/SNF components (ARID2 and SMARCE1) in CoIP westerns (Supplementary Fig. 6g, h). However, we could not detect FLI1 or LDB1 in CoIPs, suggesting that, although being proximal to PU.1, they do not interact with PU.1 directly, or that the interaction is not stable during CoIP. Although SWI/SNF components were reproducibly detected in CoIPs with wild-type PU.1, these interactions were slightly reduced with the ΔQ mutant and strongly reduced or absent with the ΔA mutant (Fig. 8e and Supplementary Fig. 6f-h).

We further confirmed that PU.1 interacts with SWI/SNF by blocking experiments and ChIP-seq of SMARCA4 (BRG1) in CTV-1 cells after induction of PU.1 or its ΔQ and ΔA mutants. Inhibition of SMARCA4 (BRG1) led to a dose-dependent

reduction of PU.1 binding across the entire genome (Supplementary Fig. 6i, j), suggesting that its inactivation may have a general effect on TF binding. ChIP-seq experiments after induction of PU.1 clearly demonstrated the recruitment of SMARCA4 (BRG1) to PU.1-remodeled sites (Fig. 8f, g). As exemplified by the *GSN* locus (Fig. 8f) and across all PU.1-binding sites (Fig. 8g), the ΔQ mutant retained some of the remodeling capacity of wild-type PU.1. However, the ΔA mutant, although retaining some of its binding capacity, was neither able to alter chromatin accessibility nor did it recruit SMARCA4 (BRG1) to the otherwise de novo-remodeled sites.

In addition, as shown in the bottom panel of Fig. 8g, the redistribution of partner TFs was disabled in the ΔA mutant. Notably, the PU.1-induced redistribution of partner factors also included SMARCA4 (BRG1), which was depleted upon PU.1 induction at the ~3 K sites that lose accessibility. Upon induction of the ΔA mutant, however, the remodeler remained at those sites.

## Discussion

The present work provides a detailed analysis of mechanisms allowing the master regulator PU.1 to shape regulatory landscapes. We show that its N-terminal AAD interacts with the SWI/SNF remodeling complex and that this interface is required for PU.1 to access and remodel chromatin de novo. However, despite its potentially strong impact on chromatin landscapes, PU.1 is not a pioneer factor in its classical definition. In vivo PU.1-binding profiles as well as in vitro binding studies suggest that PU.1 binding is constrained by chromatin and epigenetic mechanisms, in particular by nucleosome positioning. Hence, it likely initiates remodeling primarily at those binding sites (and prepares them for other factors) that it can access. Given that even classical

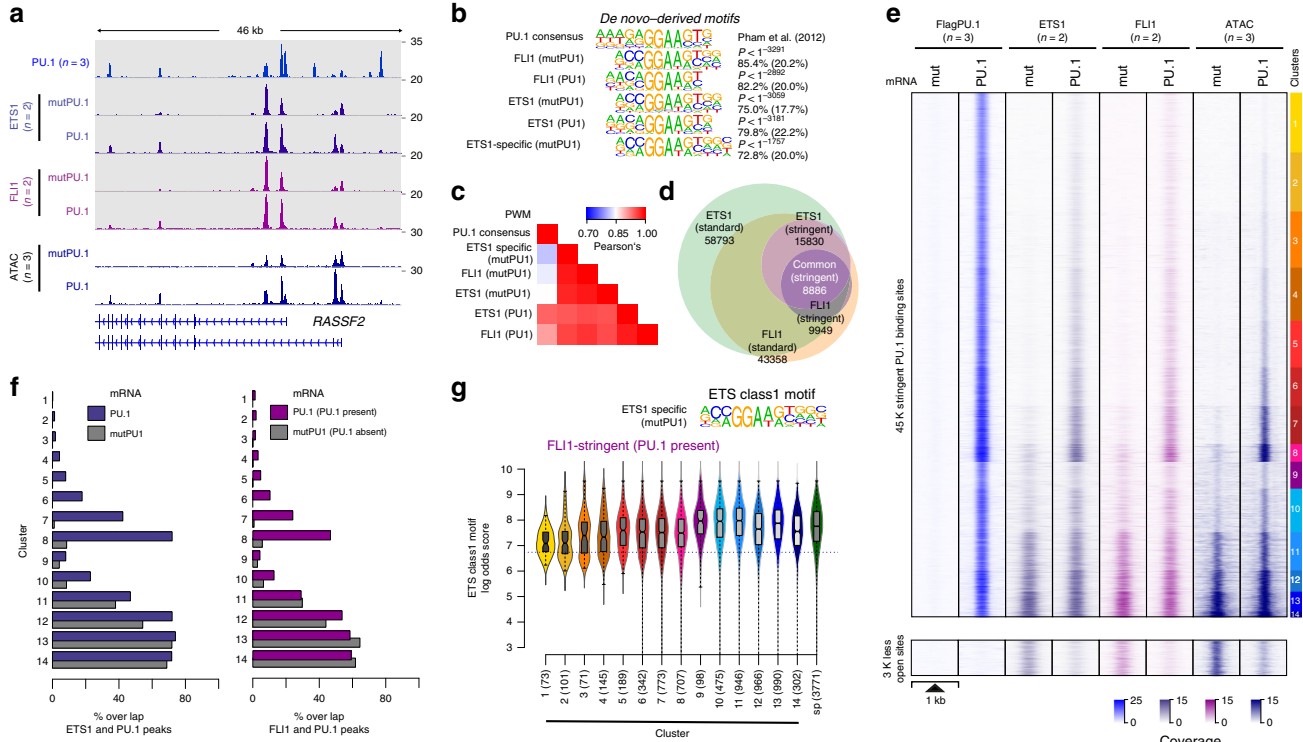

**Fig. 6 Overlap between ETS1/FLI1 and PU.1-binding sites. a** IGV genome browser tracks of the *RASSF2* locus showing average PU.1 (blue), ETS1 (purple), and FLI1 (pink) ChIP-seq coverage in control (mutPU.1) and PU.1-expressing cells. ATAC-seq coverage of PU.1-transfected and control cells are depicted in blue below the ChIP-seq tracks. **b** De novo-derived motif enrichment across the indicated ChIP-seq peaks. **c** Correlation matrix heatmap for position weight matrices (PWM) of the motifs shown in **b**. **d** Venn–Euler diagram showing the overlap of ETS1 and FLI1 ChIP-seq peaks (using stringent and standard peak calling). **e** Distribution of PU.1, ETS1, and FLI1 ChIP-seq, as well as ATAC-seq signals before and after PU.1 expression plotted across the ATAC-seq-derived PU.1 peak clusters (top panel), as well as regions that lost accessibility after PU.1 induction (bottom panel) in CTV-1 cells, as introduced in Fig. 3. **f** Bar plots displaying the overlap of stringent ETS1 (left panel) and FLI1 (right panel) peaks in PU.1-transfected (blue/purple bars) and control CTV-1 cells not expressing PU.1 (gray bars) with PU.1 peaks across the PU.1 peak clusters introduced in Fig. 3. **g** Motif log odds score distribution of the consensus ETS class 1 motif is shown for FLI1-overlapping peaks across ATAC-seq-derived PU.1 peak clusters along with FLI1 specific (sp) peaks. The median of each distribution is depicted inside the bean with a conventional boxplot. **b–g** Source data are provided as a Source Data file.

pioneer factors, which are able to bind nucleosomal target sites, depend on epigenetic and chromatin landscapes[23,24], we propose a new class of non-classical pioneer factors, such as PU.1, which share pioneering functions with classical factors but lack the ability to access nucleosome-bound target sites.

A prerequisite for pioneering is the ability to recruit remodeling complexes. Our data show that the N-terminal AAD of PU.1, which was originally defined as an activation domain using reporter assays[25] mediates the interaction with SWI/SNF. This is in concordance with earlier work demonstrating that AADs of other TFs in yeast, including VP16, Gcn4, Swi5, and Hap4, interacted directly with purified SWI/SNF complex and with the SWI/SNF complex in whole-cell extracts[26]. There is also abundant evidence for a crucial role of SWI/SNF remodeling complexes in the establishment and maintenance of lineage-specific enhancers[27–29], and their recruitment to target loci is believed to require interaction with DNA-associated TFs. Our work clearly shows that PU.1 is one of those factors that recruits SWI/SNF to its binding sites. This will allow PU.1 to fulfil its role as a master regulator and to participate in the establishment of enhancers across many hematopoietic cell types.

In a given cell type, PU.1 usually occupies a small fraction (5–10%, depending on the statistical stringency of peak calling) of its potential binding sites. Previous work indicated that PU.1-binding-site selection in individual cell types depends on its expression level and the cell type-specific mix of partner TFs[7–9]. The current work suggests that these factors only partially explain

PU.1-binding-site selection in individual cell types. In line with chromatin and epigenetic constraints observed in vitro, PU.1 cannot access all high-affinity sites in all cell types, despite its ability to recruit the remodeling machinery. The current work suggests that its pioneering role depends on cell type-specific chromatin structures, and that the ability of PU.1 to establish novel regulatory elements is likely restricted to accessible binding sites. According to our in vitro experiments, the latter may lack DNA methylation and locate to nucleosomal linker regions or sequences proximal to nucleosome entry sites. Although detailed nucleosome (and corresponding DNA methylation) maps will be required to prove this model, a restricted pioneering role of PU.1 perfectly explains the diverse and manifold PU.1 binding patterns observed across cell types. The frequent observation of shallow ChIP-seq signals (below peak detection) could be owed to the fact that nucleosome positions are not fixed across the large part of the genome and rarely synchronized across populations of cell types, which may create ample opportunities for PU.1 to bind particular recognition sequences in a (variable) sub-fraction of cells, where these sites are accessible.

Interestingly, the expression of PU.1 not only affected regulatory elements containing PU.1-binding sites. Hosokawa et al.[18] recently showed that PU.1 regulates gene expression in early T-cell development both by recruiting TFs RUNX1 and SATB1 to its own binding sites and by depleting them from the binding sites that they occupied in the absence of PU.1[18]. Our data extend their model to PU.1 partner proteins in general and implicate SWI/

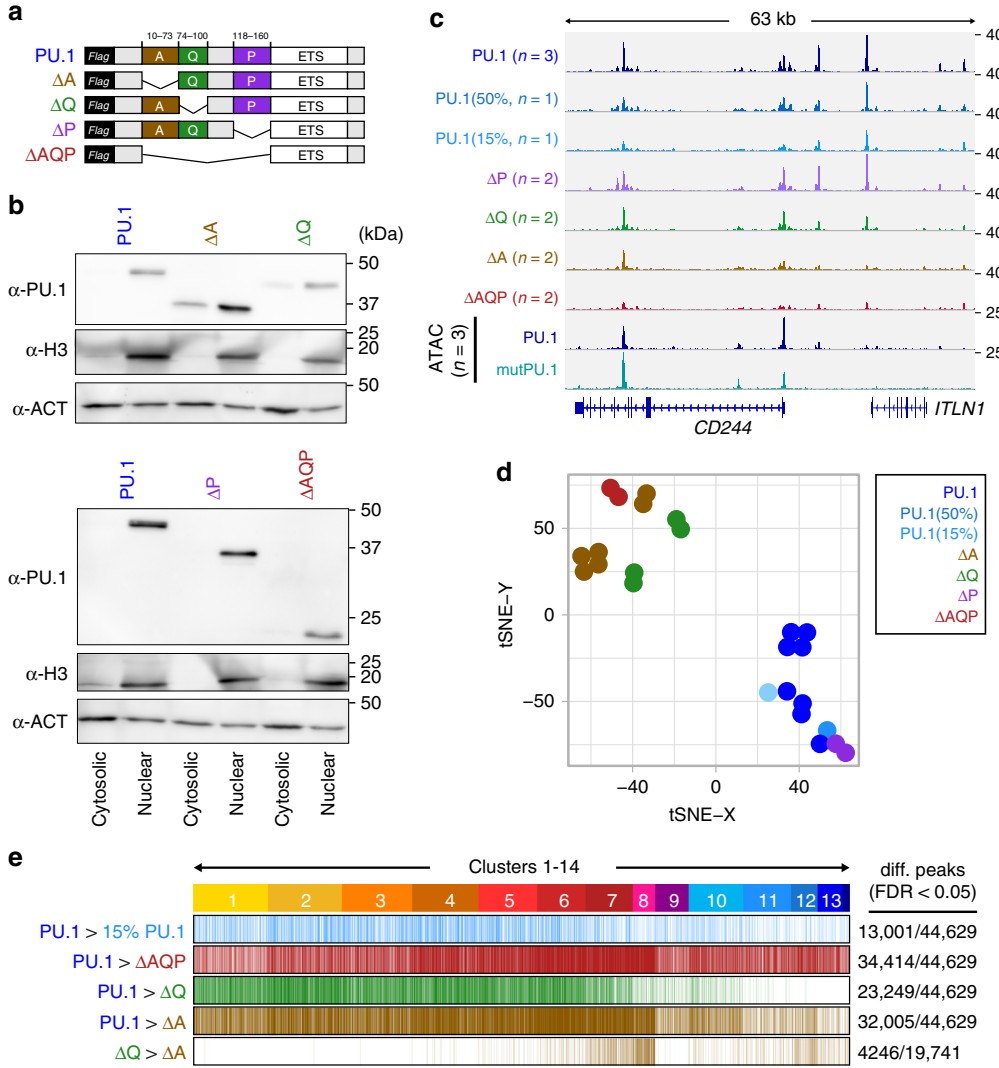

**Fig. 7 Effect of PU.1 protein domains on chromatin access. a** Design of PU.1-deletion mutants. The DNA-binding domain (ETS domain) was kept in all constructs and a 3 × -FLAG-tag was added N-terminal for detection. **b** Immunoblot confirming the nuclear expression of PU.1-deletion mutants compared with full-length PU.1 in transfected cells using IVT mRNA. **c** IGV genome browser tracks of the *CD84* locus showing PU.1 ChIP-seq coverage of CTV-1 cells transfected with varying amounts of PU.1 IVT mRNA as well as PU.1 mutant mRNA. ATAC-seq coverage of CTV-1 cells transfected with PU.1 (blue) and mutPU.1 (turquoise) IVT mRNA is depicted in the two bottom rows. **d** Two-dimensional visualization of the distribution of the indicated samples across annotated PU.1 peaks using tSNE embedding. Replicates of the same mRNA transfections are indicated by coloring. **e** Differential ChIP-seq peaks of WT vs. less PU.1 (15%, light blue), WT vs. ΔAQP (red), WT vs. ΔQ (green), and WT vs. ΔA (brown), as well as differential peaks between ΔQ vs. ΔA (brown) are shown across ATAC-seq-derived PU.1 peak clusters (introduced in Fig. 3). **b–e** Source data are provided as a Source Data file.

SNF remodelers in the process of TF redistribution. Based on our footprinting and ChIP-seq analyses, we observe the same type of factor redistribution in all three cell systems (CTV-1, TALL1, and HepG2) studied here. The PU.1-induced redirection primarily affected TFs occupying lineage-specific cis-modules, such as RUNX1 and GATA-family factors in T-cell lines and HNF- and FOXA-family factors in the liver cell line. Interestingly, the redistribution of partner TFs required the acidic domain of PU.1, which is not required for direct protein–protein interactions with RUNX1, GATA-, AP1-, or C/EBP-family factors[14–17]. Hence, the direct binding to partner proteins may not be sufficient for PU.1 to sequester partner proteins. The fact that the N-terminal acidic domain of PU.1 interacts with SWI/SNF implicates remodeling complexes in the redirection of partner proteins. The decommissioning of TF-bound cis-modules after PU.1 induction could be mediated through the reallocation of limiting SWI/SNF remodeling complexes by PU.1, which are generally required to maintain the accessible state of regulatory elements such as lineage-specific enhancers[27–30]. As many of the identified PU.1 partner proteins have already previously been shown to interact with components of SWI/SNF remodeling complexes[31–34], the latter may act as part of a hub increasing the probability of co-binding of PU.1 partner proteins at de novo-remodeled binding sites. In conclusion, our systematic analysis of de novo TF binding reveals important mechanistic details and provides more comprehensive understanding of a master regulator that shapes regulatory landscapes during hematopoiesis, has known reprogramming capabilities, but is different from "classical" pioneer factors.

## Methods
**Cell culture**. CTV-1 (DSMZ: #ACC 40), HepG2 (DSMZ: #ACC 180), K-562 (DSMZ: #ACC 10), TALL1 (DSMZ: #ACC 521), and THP-1 cells (DSMZ: #ACC 16) were grown in RPMI 1640 (Gibco) routinely supplemented with 2 mM L-glutamine

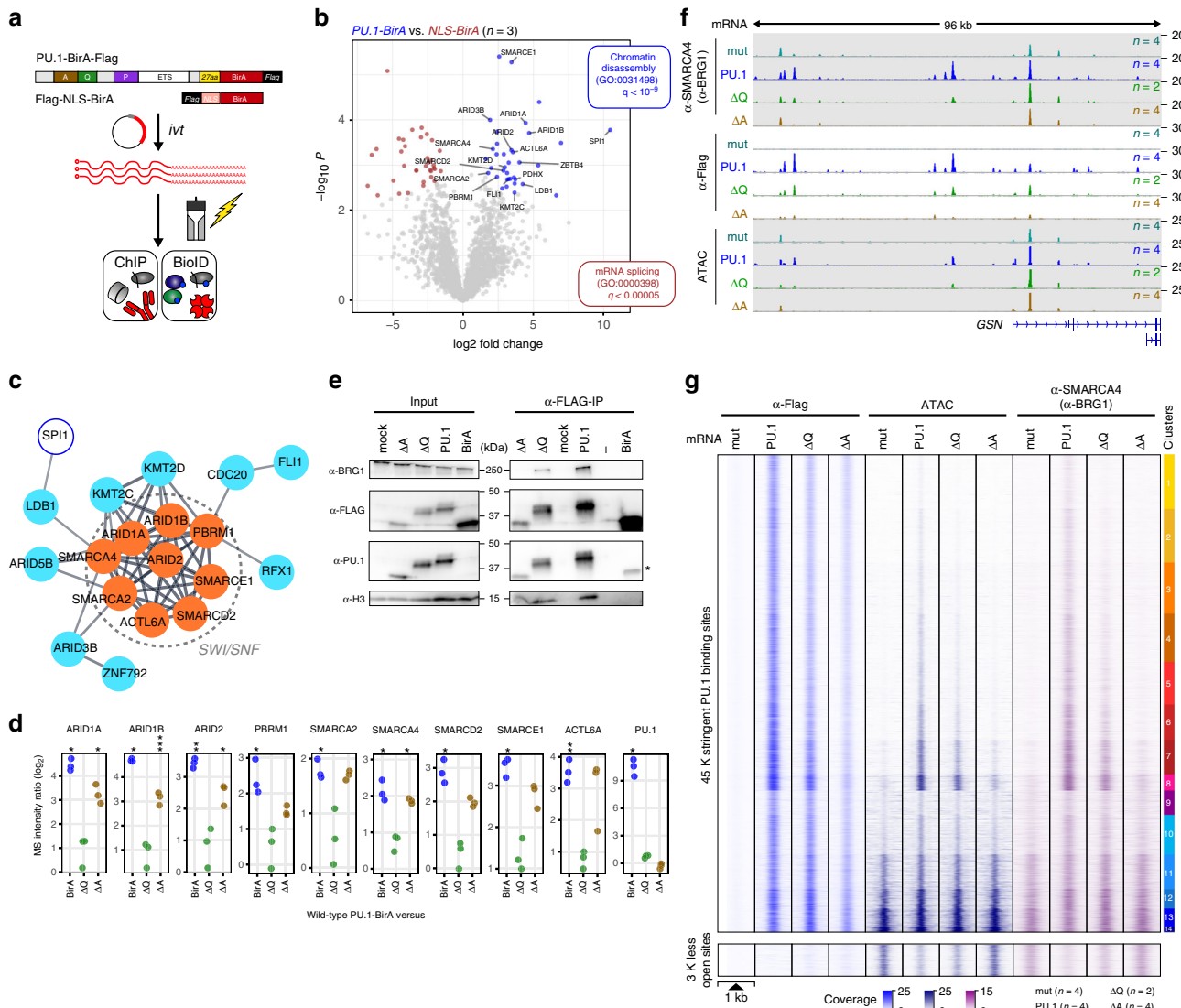

**Fig. 8 Identification of PU.1 proximal proteins using BioID. a** Schematic of the experimental setup. **b** Volcano plot illustrating proteins significantly enriched in PU.1-BirA-mRNA-transfected CTV-1 cells compared with NLS-BirA-mRNA-transfected control cells. Blue dots represent proteins with FDR < 0.05 and log fold change (logFC) > 2. Only PU.1-specific proteins of GO terms for chromatin organization and interesting transcriptional regulators are highlighted. **c** STRING analysis illustrating the functional protein association network. The network view summarizes predicted associations for proteins significantly enriched in the PU.1-BioID. The network nodes represent the proteins, the edges represent predicted functional associations. Only connected nodes are shown. **d** Dot plots showing the enrichment of peptides (ratios of $\log_2$-transformed normalized iBAQ values) representing the indicated SWI/SNF components (PU.1/*SPI1* is shown as a control) in BioIDs from PU.1-BirA vs. NLS-BirA (control), ΔQ-BirA, or ΔA-BirA-mRNA-transfected CTV-1 cells (***$P < 0.001$; **$P < 0.01$; *$P < 0.05$; paired $t$-test, permutation-based correction). **e** Immunoblotting of αFLAG mAb immunoprecipitations (5 h after electroporation, IP 5 h), along with corresponding input lysates of control (mock), PU.1, ΔQ, ΔA, and BirA-mRNA-transfected cells using the indicated antibodies. **f** IGV genome browser tracks for the *GSN* locus showing SMARCA4 (BRG1) and PU.1 ChIP-seq, as well as ATAC-seq coverage in control (mutPU.1, gray–green)-, PU.1 (blue)-, ΔQ mutant (green)-, and ΔA mutant (lightbrown)-expressing cells. **g** Distribution of the PU.1 ChIP-seq, ATAC-seq signal, and SMARCA4 (BRG1) ChIP-seq signals in control (mutPU.1), PU.1, ΔQ, and ΔA mRNA-transfected cells across the 45 K clustered PU.1-binding sites, as well as the disappearing ~3 K sites that were accessible prior to PU.1 induction. **b–e**, **g** Source data are provided as a Source Data file.

(Biochrom), 1 mM sodium pyruvate (Sigma), 50 U ml$^{-1}$ penicillin/streptomycin, 0.4× vitamins (Sigma), 1× non-essential amino acids (Sigma), 50 μM b-mercaptoethanol (Gibco), and 10% or 15% (TALL1) heat-inactivated fetal calf serum (FCS) (Gibco) at 37 °C, 5% $CO_2$, 95% humidity. Cells were split every 2–3 days and resuspended in fresh medium. Adherent HepG2 cells were detached using 1× Trypsin-EDTA (Thermo Fisher Scientific). After 20 passages, cells were discarded and a new batch of cells was used. For differentiation of THP-1 cells into MAC-like cells PMA ($10^{-8}$ M; Calbiochem) and 1,25-dihydroxyvitamin D3 (VD3, $10^{-7}$ M; Sigma) were added, cells were incubated for 3 days at 37 °C, 5% $CO_2$, 95% humidity, and adherent cells were detached using Accutase solution (Sigma). Collection of blood cells from healthy donors was performed in compliance with the Helsinki Declaration. All donors signed an informed consent. The leukapheresis procedure and subsequent purification of hematopoietic cell types were approved by the local ethical committee (reference number 12-101-0260). Separation of peripheral blood cell types and in vitro

differentiation of MOs into MACs or DCs were performed as described previously[9,35,36]. B cells were purified as described[37]. Neutrophils were purified from whole blood after lysis of erythrocytes with ACK (Ammonium-Chloride-Potassium) lysis buffer (5 ml per ml blood of 155 mM $NH_4Cl$, 100 mM $KHCO_3$, 0.1 mM EDTA) and purified by fluorescence-activated cell sorting (FACS) based on size and granularity. Human mast cells were purified from the skin that was obtained from cosmetic breast-reduction surgeries[38] with informed consent of the patients. Mast cell preparations were performed in compliance with the Helsinki Declaration and were approved by the ethics committee of the Charité Universitätsmedizin Berlin (reference number EA1/204/10).

**Transfection of in vitro transcribed mRNA.** Synthetic DNA templates (gBlocks) for wild-type PU.1, it's mutated version, all PU.1-deletion mutants, and all BirA*

fusion constructs for proximity-dependent interaction analysis were ordered at IDT (Integrated DNA Technologies). Sequences are provided in Supplementary Data File 2. Constructs were assembled into a BamHI and XbaI (NEB)-linearized T7 promoter-containing vector (pEF6; Invitrogen) either using T4 DNA ligase (Promega) or the NEBuilder HiFi DNA Assembly Master Mix (NEB). SfuI (Roche) linearized plasmids were phenol:chloroform-purified and were used to generate in vitro transcribed (IVT) mRNA with the mMESSAGE mMACHINE T7 Ultra Kit (Ambion) according to the manufacturer's instructions. IVT mRNA was purified with the RNeasy Mini Kit (Qiagen) according to the manufacturer's instructions. For electroporations, routinely $3 \times 10^6$ cells were pelleted and washed once at room temperature with phenol red-free RPMI 1640 (Gibco) and once with phenol red-free Opti-MEM I (Gibco). Cell number was scaled-up as needed. Cells were electroporated in 200 μl phenol red-free Opti-MEM I in a 4 mm cuvette using a Gene Pulser Xcell (Bio-Rad) with a rectangular pulse of 400 V and 5 ms duration, usually delivering 3 μg PU.1 IVT mRNA. IVT mRNA amounts of additional constructs were calculated according to their size relative to PU.1 (ΔA 2.3 μg, ΔQ 2.6 μg, ΔP 2.6 μg, and ΔAQP 1.5 μg of IVT mRNA). Immediately after electroporation, the cell suspension was transferred into pre-warmed culture medium ($1 \times 10^6$/ml) and cultured at 37 °C, 5% CO$_2$, and 95% humidity for the indicated times. Electroporated cells were 85–95% viable as analyzed by DAPI (4′,6-diamidino-2-phenylindole) staining and corresponding FACS analyses. Expression levels of overexpressed proteins were determined by immunoblotting 8 h after transfection.

To study the effect of long-term PU.1 expression in terms of induced gene expression changes in the heterologous CTV-1 cell line, cells were subjected to seven cycles of PU.1 and PU.1mut mRNA transfection, respectively, for 7 consecutive days prior to total RNA isolation and transcriptome analyses (Fig. 3, long).

**Co-immunoprecipitation**. PU.1 protein containing complexes were isolated by affinity purification as previously described by Hosokawa et al.[18], with slight modifications. In brief, $10 \times 10^6$ CTV-1 cells were mock transfected (electroporation without RNA) or transfected with 10 μg of PU.1 mRNA, 8.6 μg of ΔQ mRNA, 7.6 μg of ΔA mRNA, and 7.6 μg of NLS-BirA as additional control. Transfected cells were incubated for 5 h at 37 °C, 5% CO$_2$, 95% humidity prior to cell lysis using IP buffer (50 mM Tris-HCl pH 7.5, 150 mM NaCl, 10% glycerol, 0.1% Tween-20, 1 mM EDTA, 10 mM NaF, 1× protease inhibitor cocktail). Cells were solubilized on ice for 30 min with gentle shaking and sonicated three times for 10 s followed by 30 s rest on ice on a Branson Sonifier 250 (constant duty cycle, output control 2). Insoluble material was removed by centrifugation and the protein concentration of each lysate was assessed using the Qubit Protein Assay Kit. Where indicated, lysates were additionally treated with 250 U of benzonase (Sigma) for 30 min at 37 °C prior to immunoprecipitation to digest genomic DNA. Ten micrograms of each lysate were saved as input control, before a pre-clearing step with mouse IgG-Agarose (Sigma) was performed for 1 h at 4 °C on a rotating wheel. Pre-cleared protein extracts were subjected to immunoprecipitation with anti-FLAG M2 agarose (Sigma) overnight at 4 °C on a rotating wheel. Enriched complexes were eluted from the agarose using 3×FLAG peptide (Sigma) and protein concentrations were assessed using the Qubit Protein assay Kit. Ten micrograms of the immunocomplexes were separated together with the corresponding input samples by SDS-polyacrylamide gel electrophoresis (PAGE) on 10% pre-cast polyacrylamide gels (Bio-Rad). Immunoblotting was performed as described below, using the following antibodies as indicated: anti-ARID2 (sc-166117, Santa Cruz, 1:100), anti-BRG1 (ab110641, Abcam, 1:2000), anti-FLAG M2 (F3165, Sigma, 1:5000), anti-Histone H3 (ab1791, Abcam, 1:2000), anti-PU.1 (sc-352×, Santa Cruz, 1:5000), anti-SMARCE1 (ab131328, Abcam, 1:500), anti-FLI1 (ab15289, Abcam, 1:1000), and anti-LDB1 (ab96799, Abcam, 1:1000).

Reverse CoIP of BRG1-containing complexes were performed using the Dynabeads antibody coupling kit provided by Thermo Fisher Scientific. The anti-BRG1 (ab110641, Abcam) antibody was covalently coupled to Dynabeads according to the manufacturer's instructions and $10 \times 10^6$ transfected CTV-1 cells were used for the IP experiments. Cells were lysed in Dyna-IP buffer (20 mM Tris-HCl pH 7.5, 150 mM NaCl, 1% Triton X-100, 2 mM EDTA, 1× protease inhibitor cocktail) and solubilized on ice for 30 min with gentle shaking. After three rounds of sonication, 10 s each followed by 30 s rest on ice using a Branson Sonifier 250 (constant duty cycle, output control 2), insoluble material was removed by centrifugation. The protein concentration of each lysate was assessed using the Qubit Protein Assay Kit and 10 μg of each lysate was saved as input control. Lysates were incubated with anti-BRG1-coupled Dynabeads for 1.5 h at room temperature on a rotating wheel and washed three times each with phosphate-buffered saline (PBS) including Tween-20 (0.02%) and ultrapure water, respectively. For elution of protein complexes, 50 μl 2× SDS sample buffer without 2-mercaptoethanol was added to the beads, before the beads were incubated at 95 °C for 10 min. Supernatants were collected on a magnetic rack and mixed with 2.5 μl 2-mercaptoethanol each. Twenty microliters of the immunocomplexes were separated together with the corresponding input samples by SDS-PAGE on 10% pre-cast polyacrylamide gels (Bio-Rad). Immunoblotting was performed as described below, using the following antibodies as indicated: anti-BRG1 (ab110641, Abcam, 1:2000), anti-FLAG M2 (F3165, Sigma, 1:2000), anti-PU.1 (sc-352×, Santa Cruz, 1:5000), and anti-Histone H3 (ab1791, Abcam, 1:2000).

**Immunoblotting**. Immunoblotting of whole-cell extracts was performed as described previously[39] and all protein lysates were collected 8 h after transfection. Nuclear extracts were prepared using ice-cold hypotonic sucrose buffer (1% Triton X-100, 320 mM Sucrose, 10 mM Hepes pH 7.5, 5 mM MgCl$_2$, 1 mM phenylmethylsulfonyl fluoride (PMSF), 1× protease inhibitor cocktail, 1× phosphatase inhibitor cocktail). Cells were solubilized on ice with hypotonic buffer using a tissue grinder (Sigma) with ten strokes per sample. After centrifugation, the supernatant was saved as cytoplasmic fraction and cell nuclei were washed twice with ice-cold sucrose wash buffer (320 mM, 10 mM Hepes pH 7.5, 5 mM MgCl$_2$, 1 mM PMSF, 1× protease inhibitor cocktail, 1× phosphatase inhibitor cocktail). To shear the DNA, nuclei were resuspended in ice-cold nuclear sonication buffer (50 mM Tris-HCl pH 8.0, 150 mM NaCl, 1 mM EDTA pH 8.0, 10% glycerol, 1 mM PMSF, 1× protease inhibitor cocktail, 1× phosphatase inhibitor cocktail) and sonicated twice for 10 s with a rest of 30 s on ice using a Branson 250 sonifier. Sheared DNA was removed by incubation with 250 U of benzonase (Sigma) for 30 min on ice, followed by centrifugation. Nuclear extracts were transferred into fresh tubes and the protein concentration of cytoplasmic and nuclear lysates was determined using the Qubit Protein Assay Kit (Thermo Fisher Scientific). Usually, 15 μg of each fraction were incubated at 95 °C for 10 min in 2× SDS buffer, separated by SDS-PAGE and were transferred to polyvinylidene difluoride membranes (Merck). Immunoblots were probed with the following antibodies as indicated: anti-Actin (A2066, Sigma, 1:2000), anti-Histone H3 (ab1791, Abcam, 1:2000), anti-FLAG M2 (F3165, Sigma, 1:2000), or anti-PU.1 (sc-352, Santa Cruz, 1:2500).

**Gelshift assays**. Nucleosomes for nucleosome-PU.1 interaction assays were assembled as described[40]. DNA fragments for assembly were produced via PCR from gBlocks ordered from IDT. Corresponding sequences are listed in Supplementary Table 2 and PCR primer sequences are provided in Supplementary Table 3. The DNA templates consist of a stretch of D.m. HSP70 promoter followed by the 601 nucleosome-positioning sequence (NPS). The PU.1 high-affinity binding motif 5′-CACTTCCTCTTT-3′[11] was inserted at varying distances to the nucleosome border. Forward primers for DNA fragments with the PU.1-binding motif were labeled with Cy5. Forward primers for control fragments without a PU.1-binding motif were labeled with Cy3. Recombinant human PU.1 was expressed in *Escherichia coli* (Rosetta2(DE3)pLysS) with C-terminal FLAG- and N-terminal His-tags. The protein was natively purified using the His-tag on a Ni-NTA resin (Qiagen) via a gravity flow column. Protein concentration was determined via Bradford assay in comparison with bovine serum albumin (BSA) calibration solutions. Free DNA (50 nM) or nucleosomes were incubated with an excess of 4 mM recombinant PU.1 for 30 min at 30 °C in reaction buffer (20 mM Tris-HCl pH 7.6, 1.5 mM MgCl$_2$, 0.5 mM EGTA, 300 mM KCl, 10% glycerol, 10 mM dithiothreitol (DTT)). The samples were then resolved on a 0.4× TBE 6% native PAA gel for 90 min at 4 °C and 130 V. The gels were subsequently scanned on a GE FLA-9000 instrument.

**Microscale thermophoresis measurements**. Motif affinity measurements were essentially carried out as described previously[11]. In brief, binding assays were performed using annealed oligonucleotides (Cy3-labeled on one strand, listed in Supplementary Table 3) and recombinant full-length PU.1 on the Nanotemper Monolith NT.115 device. The sequence of the full-length hPU.1 was amplified by PCR from pORF9-hSPI1 (InvivoGen) and recombined into a modified pDM8 vector, encoding an N-terminal His-tag, using the Gateway technology (Life Technologies). The protein was expressed in Rosetta2(DE3)pLysS (Novagen) and purified by Nickel affinity chromatography (Qiagen) as described above. Analyzed oligomers were synthesized with Cy3-labeled, with either a methylated or hydroxyl-methylated CpG-site (5mC/5hmC), or as unmodified DNA oligomers (Sigma). For each motif, two independent sets of 16 affinity measurement reactions were prepared using a dilution series of PU.1 protein where the concentration of the double-stranded oligonucleotide was kept constant (50 nM). Data analysis was done using the NT-analysis acquisition software (1.2.229).

**ChIP-seq library preparation**. ChIP was performed in biological replicates as described previously with slight modifications[9]. Chromatin for all ChIP-seq experiments of mRNA-transfected cells was collected 8 h after transfection, besides for ChIP-seq analyses of PU.1-BirA, ΔA-BirA, and ΔQ-BirA fusion constructs, which were already collected 5 h after transfection. Briefly, for H3K27ac and PU.1/FLAG ChIP-seq, cells were crosslinked with 1% formaldehyde for 10 min at room temperature and the reaction was quenched with glycine at a final concentration of 0.125 M. For SMARCA4/BRG1, ETS1 and FLI1 ChIP-seq experiments dual crosslinking was performed. Cells were crosslinked first with 2 mM disuccinimidyl glutarate (Thermo Fisher Scientific) in PBS for 30 min at room temperature. Subsequently, cells were fixed with 1% formaldehyde for 10 min at room temperature and the reaction was quenched with glycine at a final concentration of 0.125 M.

Chromatin of all ChIP experiments was sheared using sonication (Branson Sonifier 250) to an average size of 250–500 bp. A total of 2.5 μg of antibody against H3K27ac (Abcam, ab4729), PU.1 (Santa Cruz, sc-352×), FLAG M2 (Sigma Aldrich, F3165), ETS1 (Santa Cruz, sc-350×), FLI1 (Abcam, ab15289), or SMARCA4/BRG1 (Abcam, ab110641) was added to sonicated chromatin of $2 \times 10^6$ cells and

incubated overnight at 4 °C. Protein A or G (for FLAG ChIP-seq) sepharose beads (GE healthcare) were added to the ChIP reactions and incubated for 2 h at 4 °C. Beads were washed and chromatin was eluted. After crosslink reversal, RNase A and proteinase K treatment, DNA was extracted with the Monarch PCR & DNA Cleanup kit (NEB). Sequencing libraries were prepared with the NEBNext Ultra II DNA Library Prep Kit for Illumina (NEB) according to the manufacturer's instructions. The quality of dsDNA libraries was analyzed using the High Sensitivity D1000 ScreenTape Kit (Agilent) and concentrations were assessed with the Qubit dsDNA HS Kit (Thermo Fisher Scientific). Libraries were single-end sequenced on a HiSeq3000 or NextSeq550 (Illumina). Sequencing libraries are listed in Supplementary Data File 1.

**ATAC-seq library preparation**. ATAC-seq was essentially carried out as described[41] 8 h after mRNA transfection of CTV-1, HEP-G2, and TALL1 cells. Briefly, prior to transposition the viability of the cells was assessed and $1 \times 10^6$ cells were treated in culture medium with DNase I (Sigma) at a final concentration of 200 U ml$^{-1}$ for 30 min at 37 °C. After Dnase I treatment, cells were washed twice with ice-cold PBS, and cell viability and the corresponding cell count were assessed. For each ATAC reaction, $5 \times 10^4$ cells were aliquoted into a new tube and spun down at $500 \times g$ for 5 min at 4 °C, before the supernatant was discarded completely. The cell pellet was resuspended in 50 μl of ATAC-RSB buffer (10 mM Tris-HCl pH 7.4, 10 mM NaCl, 3 mM MgCl$_2$) containing 0.1% NP-40, 0.1% Tween-20, and 1% digitonin (Promega), and was incubated on ice for 3 min to lyse the cells. Lysis was washed out with 1 ml of ATAC-RSB buffer containing 0.1% Tween-20. Nuclei were pelleted at $500 \times g$ for 10 min at 4 °C in a fixed-angle centrifuge. The supernatant was discarded carefully and the cell pellet was resuspended in 50 μl of transposition mixture (25 μl 2× tagment DNA buffer, 2.5 μl transposase (100 nM final; Illumina), 16.5 μl PBS, 0.5 μl 1% digitonin, 0.5 μl 10% Tween-20, 5 μl H$_2$O) by pipetting up and down six times. The reaction was incubated at 37 °C for 30 min with mixing (1000 r.p.m.) before the DNA was purified using the Monarch PCR & DNA Cleanup Kit (NEB) according to the manufacturer's instructions. Purified DNA was eluted in 20 μl elution buffer (EB) and 10 μl purified sample was objected to a ten-cycle PCR amplification using Nextera i7- and i5-index primers (Illumina). Purification and size selection of the amplified DNA were carried out with magnetic beads (Agencourt AMPure XP). For purification the ratio of sample to beads was set to 1:1.8, whereas for size selection the ratio was set to 1:0.55. Purified samples were eluted in 15 μl of EB. Quality and concentration of the generated ATAC libraries were analyzed using the High Sensitivity D1000 ScreenTape Kit (Agilent) and libraries were sequenced paired-end on a NextSeq550 (Illumina). Sequencing libraries are listed in Supplementary Data File 1.

**RNA-seq library preparation**. To analyze gene expression profiles of various cell types under several conditions, e.g., PU.1-transfected vs. PU.1mut-transfected cells, RNA-seq was performed 24 h after mRNA transfection (short) or after cycles of PU.1 mRNA electroporation over 7 days (long). Generation of dsDNA libraries for Illumina sequencing from total cellular RNA was carried out using the ScriptSeq Complete Kit (Illumina). Typically, 1 μg of DNA-free RNA was used for each reaction. In a first step, rRNA was depleted using the Ribo-Zero rRNA removal reagents included in the kit. rRNA-free RNA was then converted into cDNA, 3'-terminal tagged, and was used for PCR amplification and library purification. The quality of dsDNA libraries was analyzed using the High Sensitivity D1000 ScreenTape Kit (Agilent) and concentrations were assessed with the Qubit dsDNA HS Kit (Thermo Fisher Scientific). Paired-end sequencing was carried out on HiSeq1000 or HiSeq3000 instruments (Illumina). Sequencing libraries are listed in Supplementary Data File 1.

**Proximity-dependent biotinylation assay (BioID)**. BioID experiments to analyze proximity-dependent interactions of PU.1 and vincinal proteins were essentially carried out as described[22] with slight modifications to adapt the protocol to our transient mRNA transfection approach. Each experiment was performed in three independent biological replicates. Briefly, wild-type PU.1, or its ΔA and ΔQ deletion mutants, were fused to promiscuous E. coli biotin ligase (BirA*, harboring a R118G point mutation) via a flexible linker and ordered as gBlock gene fragments at IDT (sequences are given in Supplementary Data File 2). The biotin ligase fused to a nuclear localization sequence (NLS) was used as control for all experiments. Constructs were assembled into a BamHI and XbaI (NEB)-linearized T7 promoter-containing vector (pEF6; Invitrogen) using the NEBuilder HiFi DNA Assembly Master Mix (NEB). SfuI (Roche) linearized plasmids were phenol:chloroform-purified and used to generate IVT mRNA with the mMESSAGE mMACHINE T7 Ultra Kit (Ambion) according to the manufacturer's instructions. Finally, generated IVT mRNA was purified with the RNeasy Mini Kit (Qiagen) according to the manufacturer's instructions. Usually, 100 μg of PU.1-BirA* IVT mRNA was introduced into $50 \times 10^6$ cells via electroporation. IVT mRNA amounts of additional constructs were calculated according to their protein size relative to PU.1-BirA* (ΔA-BirA* 90 μg, ΔQ-BirA* 96 μg, or NLS-BirA* 50 μg of IVT mRNA). Three hours after transfection, the cell culture medium was supplemented with 50 μM biotin (Sigma) to achieve biotinylation of vicinal proteins. Five hours after biotin supplementation, cells were collected for subsequent lysis and purification of biotinylated proteins. The optimal time points for the addition of biotin

and to collect the cells were determined by immunoblotting. Cell pellets were washed twice with ice-cold PBS to eliminate residual biotin. For lysis, cell pellets were first swelled in ice-cold lysis buffer 1 A (10 mM Hepes pH 7.5, 85 mM KCl, 1 mM EDTA pH 8.0, 1 mM PMSF, 1 mM Na$_3$VO$_4$, 1× protease inhibitor cocktail), before an equal amount of ice-cold lysis buffer 1B (10 mM Hepes pH 7.5, 1% NP-40, 85 mM KCl, 1 mM EDTA pH 8.0, 1 mM PMSF, 1 mM Na$_3$VO$_4$, 1× protease inhibitor cocktail) was added. Suspensions were incubated on ice for 10 min before nuclei were spun down at $700 \times g$ for 5 min at 4 °C. In a next step, nuclei were resuspended in lysis buffer 2 (50 mM Tris-HCl pH 7.4, 0.4% SDS, 5 mM EDTA pH 8.0, 500 mM NaCl, 1 mM PMSF, 1 mM Na$_3$VO$_4$, 1× protease inhibitor cocktail) and sonicated to shear genomic DNA. All sonication steps were carried out with a constant duty cycle, output control 2, for 10 s using a Branson Sonifier 250. After the first sonication, Triton X-100 was added to a final concentration of 2% before lysates were sonified again. Finally, an equal amount of ice-cold dilution buffer (50 mM Tris-HCl pH 7.4, 1 mM PMSF, 1 mM Na$_3$VO$_4$, 1× protease inhibitor cocktail) was added and suspensions were sonified one last time. Insoluble materials were removed by centrifugation, before a dialysis step using Slide-A-Lyzer MINI Dialysis Devices (Thermo Fisher Scientific) was performed to remove residual biotin o/n at 4 °C. The next day, prewashed Pierce Streptavidin Magnetic Beads (Thermo Fisher Scientific) were added and lysates were incubated for 2 h on a rotating wheel at room temperature. Beads were washed three times each with 100 mM NH$_4$HCO$_3$ and 4 M Urea in Tris-HCl pH 8.0, respectively. On-bead digestion of captured protein complexes and subsequent mass spectrometry analysis were carried out in collaboration with the Protein Analysis Unit at the Biomedical Center (LMU, Munich, Germany). Beads were resuspended in 40 μl of 100 mM Tris pH 7.6 containing 4 M urea and 0.5 μg LysC. The mixture was incubated at 28 °C for 1.5 h to release bound proteins into the solution. The supernatant was subsequently transferred into a fresh vial and the beads were washed twice with 100 μl of 100 mM Tris pH 7.6. Both wash fractions were combined with the supernatant and DTT was added to a final concentration of 10 mM for reduction of disulfide bonds. Next, 1 μg trypsin was added to proteolytically cleave the oligopeptide mixture at 28 °C for 12 h. Subsequently, 1 M iodoacetamide in 100 mM Tris pH 7.6 was added to a final concentration of 30 mM to alkylate-free cysteine side chains. After 30 min incubation at room temperature, the samples were acidified by adding 3 μl of 90% formic acid (FA) and 10% trifluoroacetic acid. Residual beads were removed with a magnet and the oligopeptide mixture was a desalted using the C18 stage tip protocol[42]. Briefly, three layers of C18 discs (Empore C18, 3 M) were placed in a 200 μl tip, washed, and samples were loaded onto the column. For sample loading, a low centrifugation speed of $70 \times g$ was applied until the complete sample was flown through the column. For salt removal, $3 \times 50$ μl 0.1% FA were applied with fast centrifugation at $250 \times g$. Finally, peptides were eluted with $2 \times 60$ μl of 70% acetonitrile (ACN), containing 0.1% FA, subsequently vacuum dried, and reconstituted in 10 μl of 0.1% FA in water as loading buffer for the high-performance liquid chromatography (HPLC) separation.

All samples were analyzed via nano reversed-phase liquid chromatography tandem mass spectrometry (MS/MS) on the Ultimate 3000 nCS HPLC system coupled to an QExactive HF mass spectrometer. Samples were loaded onto the chromatographic column (150 × 0.075 mm, packed in-house with 2.4 μm C18 chromatographic material Reprosil-AQ, Dr Maisch GmBH) by direct injection. For peptide separation, a linear gradient over 60 min from 4% to 40% ACN in 0.1% FA was applied. The column outlet was connected to the nano-electrospray ionization source to transfer the eluting ions directly into the QExactive HF MS for peptide analysis. The mass spectrometer was operated in data-dependent acquisition mode acquiring one survey scan covering the range of 350–1600 m z$^{-1}$ at 60,000 resolution, followed by up to 10 MS/MS scans of selected peptide precursors per cycle. Suitable precursors had a defined charge state between 2 to 5+, a minimal intensity of 4.0 e4, and were isolated with in a 1.5 Da window. Peptide precursors were fragmented in the higher-energy collisional dissociation cell applying a normalized collision energy of 27 and spectra were acquired in the orbitrap at 15,000 res. To prevent repeated analysis of precursors, dynamic exclusion was programmed for 15 s with a 12 p.p.m. window around the signal of previously fragmented precursors. A list of all processed samples is provided in Supplementary Table 4.

**Detection of biotinylated proteins in whole-cell lysates**. To analyze the global biotinylation upon transfection with various mRNAs, CTV-1 cells were transfected as described above. Three hours after transfection, the transfected cells were supplemented with 50 μM biotin or not and cells were collected and lysed 5 h after biotinylation. Whole-cell lysates were separated by SDS-PAGE and were blotted as described above. Membranes were blocked in 1% BSA in PBS with 0.2% Triton X-100 and incubated in the same buffer with horseradish peroxidase-conjugated streptavidin (ab7403, Abcam, 1:10,000) overnight. After three quick washes with PBS, membranes were agitated for 5 min in 10% BSA in PBS with 1% Triton X-100 and biotinylated proteins were detected after three additional washes with PBS.

**BRG1 inhibition assay**. The small molecule inhibitor PFI-3 (Sigma) was used to selectively inhibit the SMARCA4/2 polybromo 1 domain of the SWI/SNF complex. PU.1 IVT mRNA-transfected CTV-1 cells were cultured in RPMI 1640 medium (Gibco) supplemented with 10% FCS (Gibco) containing the indicated amounts of the small molecule inhibitor PFI-3. Eight hours after transfection and

corresponding treatment, cells were collected and were used for subsequent ChIP-seq analysis as described above, using the anti-FLAG M2 antibody (Sigma) to analyze PU.1 binding patterns in the heterologous cell type upon inhibition of the SMARCA4/2 component of the SWI/SNF complex.

**RNA-seq analysis**. Sequencing reads were mapped to the human genome using STAR v2.5.3a[43]. The human hg19 genome index together with gene annotation from GENCODE[44] (release 19) was used to aid in spliced alignment. Tables of raw uniquely mapped read counts per human gene were generated during mapping using the built-in --quantMode GeneCounts option in STAR. For the comparison of gene expression levels between ETS family factors shown in Fig. 2c, relative expression data were corrected for transcript length and were plotted for selected genes using the ggplot2 (v3.1.0) package in R 3.4.3. Differential expression analysis was carried out on raw gene counts using edgeR 3.20.8[45] in R, comparing PU.1 mRNA-transfected cells with control cells transfected with a mutant PU.1 mRNA, focusing on genes with an absolute fold change > 2 and a false discovery rate (FDR) < 0.05. The volcano plot of the edgeR results in Fig. 2e was generated using the ggplot2 package in R. Statistically significant enriched GO terms were identified using Metascape[46] and the GO-term network graph in Fig. 2f was rearranged using Cytoscape[47]. To compare the expression of differentially regulated genes with the *SPI1* gene expression across various cell lineages (as shown in Fig. 2g), we extracted Cap Analysis of Gene Expression data provided by the FANTOM consortium[35] and plotted the data using the ggplot2, reshape, and ggrepel packages in R. To calculate significance levels for Pearson's correlations shown in Fig. 2g, we used the rcorr() function in the Hmisc package in R. Correlation coefficient and significance shown in Supplementary Fig. 4b were calculated using the stat_cor function in the ggpubr package in R.

**ChIP-seq analysis**. Reads (single-end) were aligned to the human genome (GRCh37/hg19) using bowtie2[48] in a very sensitive mode, keeping only reads that map to a single unique genomic location for further analysis (MAPQ > 10). Initial quality control was performed by calculating the fraction of reads in peaks (FRIP, summarized in Supplementary Table 1 and Data File 1) by running HOMER's[8] (v4.9) findPeaks program in "factor" or "histone" mode using default parameters and the appropriate matching background dataset (either ChIP input, genomic DNA, or control ChIP). For further analyses, chromosome scaffolds were removed. For cancer cell lines, we used the Control-FREEC v11.0 program[49] to determine allelic imbalances from low-depth whole genome sequencing data and corrected ChIP-seq read counts accordingly. ChIP-seq peaks were called using HOMER's findPeaks program in "factor" mode using default parameters (standard) or with -fdr 0.00001 (stringent) to identify focal peaks. Stringent peaks were further filtered for a minimal normalized tag count of 15 tags per peak. All peak sets were filtered by subtracting blacklisted genomic regions[50] and by filtering out regions with a mappability < 0.8. The latter was annotated to peak regions from mappability tracks generated with the GEM package[51] using HOMER's annotatePeaks.pl. Overlapping peak locations across multiple cell types were identified using the HOMER mergePeaks program, which was also used to count the frequency of each peak being called across cell types, and to determine unique (cell type-specific) peaks, e.g., as used in Supplementary Fig. 1b. To study the relationships between ChIP-seq samples, reads in peaks (200 bp) from individual samples were counted across merged peak sets using the HOMER annotatePeaks program. Normalization was done utilizing the rlog function of DESeq2[52]. Dimensionality reduction based on the tSNE algorithm (Fig. 7d and Supplementary Fig. 1a) was done using the Rtsne package and visualized using the ggplot2 package in R. The heatmap of normalized and scaled read counts across unique peaks (Supplementary Fig. 1b) was plotted in R.

To structure the PU.1 peak sets obtained after PU.1 mRNA transfection based on chromatin accessibility, we used the kmeans function in R to cluster ATAC read counts in 300 bp peak-centered windows of PU.1-transfected and control-transfected cells. Clusters were ordered to reflect increasing accessibility (as shown in Fig. 3). To assign genes to peaks in individual clusters, we identified associations in a stepwise process. First, TSSs (transcription start sites; derived from GENCODE release 19 transcripts) for expressed genes (as determined from RNA-seq data) were determined. Second, every gene was assigned to a regulatory domain as follows: each gene is assigned a basal regulatory domain of 25 kb upstream and downstream of the TSS (regardless of other nearby genes). Next, the gene regulatory domain is extended in both directions to the nearest gene's basal domain but no more than 250 kb in one direction. Then, each genomic region is associated either with a gene, if it is close to its promoter (within 1000 bp distance to a TSS), or if promoter distal, the region is checked for overlaps with GTEx expression quantitative trait loci (eQTL) SNPs (single-nucleotide polymorphisms; from whole blood, release 7). When within a 1000 bp distance to an eQTL, all genes associated with the SNP are assigned to this region. If there is no overlap with either TSS or eQTL, regions are associated with all genes whose regulatory domain they overlaps. To determine whether the expression of genes associated with individual peak clusters was significantly upregulated, we compared their RNA-seq gene expression data in R using a paired Wilcoxon's test (as indicated in Fig. 3).

Read coverage across individual peaks sets (as shown in Figs. 3, 6e, 8g or Supplementary Fig. 3b) was calculated using HOMER's annotatePeaks.pl with parameters "-hist 25 -ghist" using merged replicate ChIP-seq data sets and plotted in R using the image function.

Positions of PU.1 peaks relative to genes (Fig. 4e) were extracted using HOMER's annotatePeaks program based on gene annotation from GENCODE (release 19). The corresponding stacked bar chart was generated in R using the ggplot2 package.

Statistically significant differences in read counts across peaks between sets of ChIP-seq experiments were determined using HOMER's getDifferentialPeaksReplicates.pl, which utilizes statistical modeling functions of DESeq2. ChIP-seq coverage across peak sets was determined using the -hist option of HOMER's annotatePeaks.pl and "-hist 25 -ghist". Average coverage data and 95% confidence intervals were then calculated in R and the ggplot2 package was used to draw histograms (Supplementary Figs. 4e, f, 5a, b, d, h, and 6j).

To generate scatter plots comparing ChIP-seq data for BirA- and control constructs across a common peak set (Supplementary Fig. 6c), reads were counted using HOMER's annotatePeaks program. Scatter plots were drawn in R using the ggplot2 package and corresponding correlation coefficients were calculated in R.

Overlaps between PU.1 and ETS factor peaks (Fig. 6f and Supplementary Fig. 5c) were determined using bedtools' intersect program[53] and corresponding bar charts of peak fractions were drawn in R using the barplot function.

Plots illustrating the abundance of differential peak sets across clustered peaks (Fig. 7e) were generated in R using the image function.

**ATAC-seq analysis**. Reads (paired-end) were aligned to the human genome (GRCh37/hg19) using bowtie2 in very sensitive and no-discordant modes, keeping only reads that map to a single unique genomic location for further analysis (MAPQ > 10). Read positions were adjusted to move the ends proximal to the Tn5-binding site (for reads on the positive strand, the start is shifted +4 bp and its partner reads start −5 bp; for reads on the negative strand, the start is shifted −5 bp and its partner reads start +4 bp). Initial quality control was performed by calculating the FRIP (summarized in Supplementary Data File 1) by running HOMER's findPeaks program in using parameters "-region -size 150". We used the Control-FREEC 11.0 program to determine allelic imbalances from low-depth whole genome sequencing data and corrected ATAC-seq read counts accordingly. ATAC-seq peak regions were called by combining two different approaches: the basic peak region set was called using HOMER's findPeaks program in "region" mode using parameters "-size 150 -minDist 250 -L 2 -fdr 0.00001" to identify regions of variable length by stitching nucleosome-size peaks. To exclude shallow peak regions, only those were kept that overlapped a second peak set that was generated in "factor" mode using parameters "-size 250 -minDist 250 -L 2 -fdr 0.00001" to identify focal peaks. Statistically significant differences in read counts across peaks between sets of ATAC-seq experiments were determined using HOMER's getDifferentialPeaksReplicates.pl.

Read coverage across individual peaks sets (as shown in Figs. 3, 6e, 8g or Supplementary Fig. 3b) was calculated using HOMER's annotatePeaks.pl with parameters "-hist 25 -ghist" using merged replicate ATAC-seq data sets and plotted in R using the image function.

**Motif analysis**. The consensus motif for PU.1 was originally described in Pham et al.[9]. To determine all 12-mers across the genome overlapping the PU.1 consensus sequence, we used HOMER's[8] scanMotifGenomeWide program. All motif locations were then filtered for blacklisted genomic regions[50], extended to 200 bp regions, and filtered for mappability, equivalent to ChIP-seq peaks. Overlaps between motif-matching sequences and peak regions were determined using the bedtools' intersect program. Motif regions with no evidence of ChIP-seq signal (unbound) were determined by annotating normalized PU.1 ChIP-seq reads across all available natively PU.1-expressing cell types using HOMER's annotatePeaks.pl and those with read counts > 3 in any sample were removed. Venn diagrams for overlapping motif or peak region sets (Figs. 1b and 6d) were drawn in R using the venneuler package. Motif log odds scores for PU.1 motif- or other motif-overlapping sequences in motif or peak region sets were calculated using HOMER's annotatePeaks program. Distributions of motif scores (as shown in Figs. 1c, 4a, and 6g, and Supplementary Figs. 1b, e, 3e, and 5e) were visualized using the beanplot package in R.

De novo motif discovery in peaks or regions (e.g., as shown in Figs. 4b, 5a, and 6b, and Supplementary Figs. 2b, 3c, d, h, j, and 5a, b) was performed with HOMER's findMotifsGenome program and parameters "-len 7,8,9,10,11,12,13,14 -h". For searches in ChIP-seq peaks we used a 200 bp peak-centered window, whereas for differential ATAC regions the given region sizes were used. De novo motifs were further filtered using HOMER's compareMotifs.pl and parameters "-reduceThresh .75 -matchThresh .6 -pvalue 1e-12 -info 1.5". For motif searches across all ATAC peak regions, we reduced the search space by focussing on Tn5 integration sites. Here, "small" peaks were called in "region" mode using parameters "-size 12 -fragLength 1 -minDist 16 -L 0". These small focussed ATAC regions were then intersected with the original ATAC region set, extended by 48 bp on each side, merged and finally reduced by 24 bp on each side using the program suite bedtools.

To compare motif enrichment across different cell types or peak clusters, de novo motifs derived from unique or clustered peak sets were combined and filtered using HOMER's compareMotifs program and parameters "-reduceThresh .75

-matchThresh .6 -pvalue 1e-12 -info 1.5", keeping only motifs that matched known motifs with a correlation > 0.85. Filtered de novo motifs (e.g., as in Fig. 4b) were then annotated back to each peak set using HOMER's findMotifsGenome.pl using parameters "-size 200 -mknown -nomotif -h", and results were visualized in R using the ggballoonplot function of the ggplot2 package where balloon size corresponds to motif enrichment and corresponding *P* values are color coded (as in Supplementary Figs. 1c, 2c, and 3c, d).

To determine peak-wise motif co-association we first performed a known motif search using HOMER's findMotifsGenome.pl across unique or clustered peak sets with the PU.1 motif masked and determined the list of known motifs overlapping the previously determined de novo motif classes (e.g., Ebox, GATA, or RUNX). All listed motifs (except PU.1) were then counted in peak regions using HOMER's annotatePeaks.pl with parameters "-m *known.motifs* -fm *PU1motif* -matrixMinDist 4 -nogene -noann -nmotifs". Motif overlap in each individual peak was then reduced to motif class overlap (using the filtered known motif list), which was counted as positive for a particular class, if one of the class matching known motifs was present, or negative, if none was present. To count PU.1 motifs in peaks, we used HOMER's annotatePeaks program with parameters "-m *PU1motif* -matrixMinDist 6 -nogene -noann -nmotifs". The combined count table was then used to generate a motif co-occurrence matrix and to calculate node sizes and edges width (each represented as % of all peaks). Networks of motif co-association were generated in R using the igraph package (as shown in Fig. 4c and Supplementary Fig. 1d). To improve the visualization, colors of individual nodes were edited in Adobe Illustrator.

Footprints across motif-centered peak sets (centered using HOMER's annotatePeaks.pl programm) were generated using the -hist option of HOMER's annotatePeaks.pl with parameters "-hist 1 -len 1" and plotted in R using the ggplot2 package (as shown in Figs. 4g and 5b–d, and Supplementary Fig. 3i, k).

To study the evolutionary conservation of PU.1 motifs in clusters, PhastCons scores were annotated to PU.1 motif-centered cluster peaks, as well as matching random unbound control motifs with HOMER's annotatePeaks program using the hg19.100way.phastCons.bw file from the UCSC Genome browser. To reduce bias for low-affinity motifs, matching random unbound control motifs were generated for each of the 14 Kmeans clusters by extracting the frequency of each nucleotide sequence ("word") in each cluster using the homerTools program and randomly selecting the same number (if available) of each word from the set of motif regions with no evidence of ChIP-seq signal (defined as max. normalized read count of one in a motif-centered 200 bp window).

To study the relationship between DNA methylation at the GGAA-proximal CpG (CGGAA), which was shown to inhibit PU.1 binding in microscale thermophoresis assays, we utilized published DNA methylation data for HepG2 cells (GEO acc. GSM1204463). We extracted positions of all sequences covered by the consensus PU.1 motif that contained a GGAA-proximal CpG motif (20846 of $1.88 \times 10^6$ mappability-filtered PU.1 recognition sequences) using homerTools' extract function, removed paired motifs (were ChIP-seq signals may derive from a neighboring PU.1 motif lacking the proximal CpG), and annotated the remaining sequences (or the subset overlapping stringent HepG2 peaks) with the available DNA methylation data. Distributions of DNA methylation ratios (as shown in Supplementary Fig. 3g) were visualized using the beanplot package in R.

Homotypic PU.1 motif pairs with a max. distance of 150 bp were initially determined from all motif occurences, keeping orientation, distance and motif scores of each motif in each pairing. To determine whether there is a preferred distance/orientation of motifs, pairs were overlapped with PU.1 peaks to separate pairs into bound and unbound fractions, which were further divided into pairs with motifs in the same (sense-sense) or opposite orientations (sense–antisense). Significant enrichment (hypergeometric $P < 0.05$) of a particular motif distance was calculated in R using the phyper function. Distance frequencies of bound and unbound pairs in the same or opposite orientations were plotted in R using the ggplot package and significantly enriched distances were marked by orange color (Supplementary Fig. 4a). As the range of 12–50 bp was significantly overrepresented in the bound fraction, further analyses were focused on pairs in this range. Non-paired (single) motifs were defined as lacking a neighboring PU.1 motif within a 150 bp distance. Paired or single motif region sets were further divided into ETS-bound or ETS non-bound motifs using bedtools' intersect program. Plots illustrating the abundance of paired or single motifs across clustered peaks (Supplementary Figs. 4d and 5f) were generated in R using the image function.

**Generation of read coverage tracks**. HOMER was used to generate sequencing-depth normalized bedGraph/bigWig files of ChIP-seq and ATAC-seq data (using standard parameters for ChIP and a fixed fragment length of 65 bp for ATAC). BedGraph/bigWig files of RNA-seq data were generated during alignment using the "--outWigType bedGraph" option of STAR. BigWigs from replicate data sets were averaged using the program bigWigMerge[54] and dividing the count data by the number of sample. Resulting bedGraph files were converted to BigWig using the program bedGraphToBigWig[54]. Tracks were visualized using the IGV browser[55]. Selected regions (as shown in Figs. 1a, 2d, 6a, 7c, and 8f, and Supplementary Figs. 3a and 6b, i) were exported in svg format and formatted in Adobe Illustrator.

**Binding site prediction**. To predict which PU.1 motifs are bound in a given cell type, we modeled the probability of a peak (as defined using HOMER's findPeaks program in "factor" mode with parameter "-fdr 0.00001" to focus on stringent peaks) by multiple logistic models with different sets of predictor variables. These included motif log odds scores (annotated using HOMER's annotatePeaks.pl program with parameters "-m *PU1motif* -mscore"), conservation of motifs provided as average PhastCons scores (annotated with HOMER's annotatePeaks.pl program using the hg19.100way.phastCons.bw file from the UCSC Genome browser), chromatin accessibility before and after PU.1 induction (represented by ATAC-seq signals annotated into 50 bp, motif-centered regions using HOMER's annotatePeaks.pl program with parameter -len 1), and neighboring motifs of putative co-associated factors (initially determined by de novo motif searches in ATAC-seq regions of control cells performed with HOMER's findMotifsGenome program and parameters "-len 7,8,9,10,11,12,13,14 -h", filtered using HOMER's compareMotifs program and parameters "-reduceThresh .75 -matchThresh .6 -pvalue 1e-12 -info 1.5", keeping only motifs that matched known motifs with a correlation > 0.85). Neighboring motifs were represented by binary variables, indicating their presence between 6 bp and 100 bp from the center of the PU.1 motif. Higher-order interaction terms of these variables were not found to improve predictivity.

We trained the logistic models on a randomly selected half of all candidate sites and evaluated their predictivity on the remaining half by their receiver operating characteristic (ROC) curves and corresponding areas under the curve using the pROC package in R, as shown in Fig. 4f and Supplementary Fig. 3f.

**MS data analysis**. MS data were searched in the MaxQuant software suite (1.6.0.16) against the human database (Uniprot, 02/2015) using the andromeda search algorithm and finally filtered for a FDR of 5% on protein level and 2% on the level of peptide-to-spectrum matches. The iBAQ quantification option was enabled and peptide modifications of methionine (oxidation, variable), cysteine (carbamidomethylation, fixed), and protein N-terminus (acetylation) were selected. For further statistical analysis of the proteomics data, iBAQ values were log2 transformed, median normalized, and all proteins with less than two valid values in all experimental conditions were filtered out. Missing values in the residual dataset were filled up with random values from a Gaussian distribution with downshift of 1.9 and width of 0.3 to simulate noise using the Perseus software suite (version 1.5.5.3 for THP-1 and 1.5.8.2 for K-562 and CTV-1 cells). Subsequently, experimental groups of background control (NLS-BirA) and TF BioIDs (PU.1-BirA, ΔQ-BirA, or ΔA-BirA) were compared via two-sided Student's *t*-test with a permutation-based FDR rate correction of 5%. For this comparison, the Null hypothesis s0 was set to 0.5 and the number of permutations was set to 250. Statistically enriched proteins were further subjected to GO-term analysis using the gene annotation and analysis resource Metascape. Protein networks shown in Fig. 8c were visualized using the STRING network app in Cytoscape 3.6.1. The volcano plot (Fig. 8b) was drawn in R using ggplot2 and ggrepel packages. Dot plots shown in in Fig. 8d and Supplementary Fig. 6d were drawn in R using ggplot2 and grid packages.

**Reporting summary**. Further information on research design is available in the Nature Research Reporting Summary linked to this article.

## Data availability

The GEO accession number for the NGS raw data and processed data files (bigwig tracks, peak files) is GSE128837. The mass spectrometry proteomics data have been deposited to the ProteomeXchange Consortium via the PRIDE[56] partner repository with the dataset identifier PXD013167. All other relevant data supporting the key findings of this study are available within the article and its Supplementary Information files or from the corresponding author upon reasonable request. The source data underlying Figs. 1a–c, 2b, c, e–g, 3, 4a–g, 5a–d, 6b–g, 7b–e, and 8b–e, g, and Supplementary Figs. 1a–d, 2a–c, 3b–k, 4a-g, 5b–h, and 6a, c–h, j are provided as a Source Data file. A reporting summary for this article is available as a Supplementary Information file.

## Code availability

No new analysis software was developed for this study. Code required to reproduce the results discussed herein are available upon request.

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

## Acknowledgements

We thank Rüdiger Eder and Jaqueline Dirmeier for their support in FACS analyses, Simone Thomas for sharing methods, Sven Heinz for sharing methods and commenting on the manuscript, and Christopher Benner for comments on the manuscript. This study was funded by grants of the DFG to M.R. (RE 1310/17) and G.L. (LA 1331/17). ChIP- and RNA-sequencing were conducted at the biomedical sequencing facility (BSF) of the CeMM (Vienna; Austria), at the KFB (Kompetenzzentrum Fluoreszente Bioanalytik; Regensburg; Germany) and the NGS Core of the Regensburg Center for Interventional Immunology (RCI, University Regensburg and University Medical Center Regensburg, Germany).

## Author contributions

M.R. designed the study with additional input from G.L., J.M. and A.I. J.M. performed most experiments with contributions from J.R., C.S., C.G., K.M., A.R., S.S., D.G. and

M.N. A.F. performed in vitro binding studies on nucleosome-bound DNA. A.S. and A.I. performed mass spectrometry and data analysis. M.B. provided mast cells. M.E. and P.H. performed cell sorting. M.R. analysed sequencing data with help from J.M. Binding site predictions were done by R. Schill and R. Spang. M.R. and J.M. wrote the original draft with contributions from all authors.

## Competing interests

The authors declare no competing interests.
