## [Peer Review File · Nature Communications]

Reviewers' comments:

Reviewer #1 (Remarks to the Author):

The authors aim to determine the features that govern PU.1 activity in cells and in vitro. This is a comprehensive dissection of PU.1 that while extensive, doesn't really bring any significant new thinking to the field, unfortunately. This is certainly true for Fig1 (as has been done for other ETS factors as well) and Fig2 (that PU.1 induces myeloid exp programs). Figure 3 discusses accessibility clusters (which are just presented but not analyzed), Figure 4 presents analysis of overlap with PU.1 (SPI1) and ETS factors—again, expected and this can be gleaned from previously published genomic datasets. In Fig 5, very minimal changes are detected with respect to accessibility upon interrogation of the PU.1 mutations shown (especially on example tracks). Finally, while the authors try to identify proximal proteins using BioID and other strategies, they do not validate any of them well or using robust methodology. For example, to evaluate binding to the well-studied SWI/SNF complex (for which they pull out several members), they should have used antibodies toward a number of components, not only BRG1 (in the IP-western blot experiments). They should also validate the other proteins recovered so that the dataset can be evaluated in terms of utility. Further, the ATAC-seq and ChIP-seq for BRG1 signals are quite low and the key "points" of the figure are altogether missed. Overall, this study is more well-suited to a lower-ranked journal such as PNAS or PLOS.

Reviewer #2 (Remarks to the Author):

The study of Minderjahn and colleagues investigates how PU.1, a major regulator of hematopoiesis, accesses DNA and re-models adjacent chromatin. They use a combination of analysis of existing and new ChIPseq data to examine PU.1 binding across of swathe of human blood cell lineages and transformed cells, as well as an ectopic expression system. These approaches allow the authors to make some general conclusions about the factors that affect PU.1 DNA binding and transcription regulation. Finally, the authors use Mass Spec approaches to identify SWI/SNF the re-modelling complex that interacts with PU.1 to remodel chromatin at target sites.

Overall this is a thorough and interesting manuscript, that uses some nice approaches to extract useful information out of the global DNA binding data. The data cover some of the same territory as others including Heinz (ref6) and Ghisletti (ref7) and the same authors (ref8, 10), but I feel that the present study clearly extends that earlier work in some important new directions. I feel that this manuscript would be suited for nature Comms with some relatively simple improvements.

1. Concentration of PU.1 in the mRNA transfection model. There is abundant literature to indicate that PU.1 function is concentration dependent, yet the western in figure 2A doesn't show any reference data of endogenous PU.1-expression in other cell lines. I realise that this is not a perfect way of quantifying relative expression, but at least it will give a rough estimate as to whether the expression is low (as in B cells), high (macs) or supraphysiological, as this may impact on some site usage. An alternative quantitation approach could use the relative mRNA levels in RNAseq datasets.

2. Following on from above, given the impressive number of samples amassed in figure 1A, it would be useful to provide analysis of the numbers of ChIPseq peaks in each cell type and ideally how these correlates (or not) with PU.1 mRNA (or cell type).

3. Can the authors clarify what type of cell, CTV-1 cells are representative of. The manuscript is very unclear on this, as is the literature, but as the cells do not express PU.1 it would appear to me that the cells are likely to be of T cell origin.

4. Figure 1B. Can the authors clarify in the legend how “no signal” (red) differs from the gray “mappable” area (contains PU.1 consensus site, but no PU.1 bound). This is unclear to me.

5. Figure 4F, S5C. Again, I am struggling to understand this graph. Particularly what the % overlap of the mutPU.1 (gray) samples indicates. Could the authors clarify this in the legend?

Reviewer #3 (Remarks to the Author):

This study by Minderjahn and colleagues investigates the property of PU.1, a major transcription factor, in binding and remodeling the chromatin to eventually modify its accessibility. They use a combination of high-throughput genome-wide analyses to investigate the mechanisms by which PU.1 occupies its DNA binding sites and shapes regulatory landscapes. The authors conclude that the PU.1 N-terminal domain binds to the SWI/SNF chromatin remodeling complex and that this interaction acts as a remodeler to control the distribution of transcription factors onto chromatin.

This is an interesting study. The paper is well written and the in vitro analysis of PU.1 access to chromatin is convincing and technically sound. However, although the link between PU.1 and the swi/snf complex is quite appealing and novel, some of the presented data lack of clarity and additional experiments are necessary to allow all the drawing of conclusions as stated in the manuscript. While I understand that it might not be feasible for the authors to address during the time of the revision all of my concerns, attention to the following comments would further improve the manuscript:

Major points:

The author show that PU.1 interacts with the swi/snf complex. They further suggest that (1) this interaction is mediated by the N-terminal AAD domains of PU.1, and (2) this interaction occurs on chromatin.

1) Does PU.1 directly bind to the swi/snf complex? This is suggested by the Flag IP in Fig 6D but should be confirmed by additional experiments.

Do reverse IPs with BRG1 and/or other members of the swi/snf complex pull down PU.1 protein? Is the PU.1 - swi/snf complex interaction dependent on DNA or is it a direct protein-protein interaction? In case of a direct protein-protein interaction, is it mediated solely by BRG1 - PU.1 binding?

2) In Fig 6, can authors follow up on the mass spectrometry data by blotting for other members of the swi/snf complex than BRG1 upon Flag-PU.1 immunoprecipitation. PU.1 binding to the swi/snf complex is solely revealed by Pu.1-BRG1 interaction throughout the manuscript.

Why did the author have decided to follow up with BRG1?

Authors claim that the PU.1 - swi/snf complex interaction is not cell line dependent. However; it does not seem to be the case in K562 cells where PU.1 does not bind BRG1 as shown in the volcano plot in Fig S6D. Does PU.1 - BRG1 interaction significantly enrich the K562 cells.

3) Authors claim that the N-terminal AAD domains of PU.1 interact with the swi/snf complex. This is mainly supported by the fact that the Δ A PU.1 fragment does not bind to BRG1 or to other members of the swi/snf complex (Fig 6D and S6D). They further claim that Δ A mutant lacks remodeling capacity (Fig 5E).

Assuming that PU.1 interacts with swi/snf on the chromatin, it would be important to assess the chromatin binding capability of the Δ A PU.1 fragment. If Δ A binds to chromatin, does it lack the

interaction with BRG1 on chromatin?

In addition, can authors show the volcano plots in S6D for ΔA PU.1 vs NLS birA.

The volcano plot for the ΔQ mutant is missing as well.

In addition, author should show in Fig 6D that ΔQ mutant behaves like the WT PU.1 protein and bind BRG1.

4) In the abstract, author say "PU.1's N-terminal acidic activation domain and its ability to recruit SWI/SNF remodeling complexes". However, this statement is not supported by experiments.

Authors should assess the swi/snf complex recruitment to the chromatin upon inhibition of PU.1.

The reciprocal experiments shown in S6E-F does not directly support their claim.

5) Regarding the BirA experiments, can the author provide with westernblot for global biotination with and without the 50 uM biotin treatment for all cell lines used for mass spectrometry.

Minor points:

1) PU.1 Ab is required in Fig 6D.

2) In Fig 6B, can the author explain if log2 fold change refers to ibaq or intensity?

3) In Fig 6B, SMARCA2 labeling is misleading. The protein seems depleted when it is enrich in PU.1 immunoprecipitated samples

Response to reviewer's comments:

We would like to thank the reviewers for the time and effort they have invested in reviewing our manuscript. In response to their valuable comments and recommendations, we have acquired additional data to further support our findings, including supplemental ChIP-seq and ATAC-seq experiments related to the ΔQ deletion and many additional CoIP experiments corroborating the interaction between SWI/SNF components and PU.1, as well as the loss of interaction with the ΔA deletion.

All changes in the main or supplemental text are indicated by red lettering. Changes in figures include:

- Figure 4A: replaced with higher resolution/altered coverage range of ChIP-seq tracks to make image clearer
 - Figure 4B: added additional labelling for clarity
 - Figure 5B: replaced Western blots with higher resolution images
 - Figure 5C: replaced with alternative example region with higher resolution of ChIP-seq tracks to make image clearer
 - Figure 5D: regenerated tSNE plot including 6 novel data sets generated for this revision
 - Figure 6B: regenerated volcano plot using revised statistical analysis
 - Figure 6C: regenerated STRING network based on revised statistical analysis
 - Figure 6D: revised representation of MS-data
 - Figure 6E: new CoIP Westerns for Brg1, including the ΔQ mutant and α -PU.1 stainings
 - Figure 6F: replaced with alternative example region and including additional ChIP/ATAC-seq tracks for the ΔQ mutant and additional replicates for all other tracks
 - Figure 6G: replaced with image including additional ChIP/ATAC-seq tracks for the ΔQ mutant and additional replicates of other samples
-
- Figure S1A: regenerated tSNE plot to generate embedding for source file
 - Figure S2A: new Western blot for PU.1 in whole cell lysates of PU.1 expressing cell types compared to mRNA transfected CTV-1
 - Figure S5C: added additional labelling for clarity
 - Figure S6A: new Western blot for biotin in whole cell lysates of mRNA transfected CTV1
 - Figure S6B: resized image
 - Figure S6C: corrected correlation coefficients (which were wrong in the original figure)
 - Figure S6D: revised representation of MS-data
 - Figure S6E: new CoIP Westerns for Brg1, including benzonase treatment
 - Figure S6F: new reverse CoIP Westerns for PU.1 (α -BRG1 IP), including the ΔQ mutant and α -PU.1 stainings
 - Figure S6G: new CoIP Westerns for ARID2 (α -FLAG IP), including the ΔQ mutant and α -PU.1 stainings
 - Figure S6G: new CoIP Westerns for SMARCE1 (α -FLAG IP), including the ΔQ mutant and α -PU.1 stainings

We feel that the new data significantly strengthen our conclusions. We hope the reviewers now agree that our findings make a significant and important contribution to our understanding of how PU.1 operates.

Point-by-point response:

Reviewer #1 (Remarks to the Author):

The authors aim to determine the features that govern PU.1 activity in cells and in vitro. This is a comprehensive dissection of PU.1 that while extensive, doesn't really bring any significant new thinking to the field, unfortunately. This is certainly true for Fig1 (as has been done for other ETS factors as well)

This comment suggest that findings related to any ETS factor can be generalized. However, it is clear from their structures, expression profiles and published functional studies that they are non-redundant, interact with different proteins and serve different functions. PU.1 has a

specific and important function to the hematopoietic system and the presented work sheds new light on how it is able to shape chromatin landscapes in immune cells. The collected data is broad and unique and none of our findings can be derived from existing data of other ETS factors as suggested by the reviewer.

... and Fig2 (that PU.1 induces myeloid exp programs).

We don't claim that this is a new finding. However, since we are using a transient mRNA expression protocol, which has not been utilized in this context, we feel that these experiments and their analysis are required to demonstrate the validity of our approach.

Figure 3 discusses accessibility clusters (which are just presented but not analyzed)

We present analyses of accessibility clusters in Figures 3A-H, as well as in Figure S2A and S2B. This covers the association of clusters with induced gene expression (in 3A) comprehensive analyses of associated PU.1 motif properties (3B, 3D, 3H, S2A), composition of co-associated motifs (in 3D, S2B) and genome ontology per cluster. It is unclear to us, what the reviewer is missing here.

Figure 4 presents analysis of overlap with PU.1 (SPI1) and ETS factors—again, expected and this can be gleaned from previously published genomic datasets.

Other studies have indeed investigated the overlap of ETS-factor binding in other systems. However, this figure deals with the genomic redistribution of ETS factors immediately upon PU.1 induction. We apologize that we didn't illustrate this properly and have now rephrased the figure legend and graphs to make this clearer. The induction of PU.1 massively changes the pattern of ETS1/FLI1 distribution and allows ETS1/FLI1 binding to many novel sites, which further highlights the unique ability of PU.1 to access binding sites. We are not aware of any published data that would address the redistribution of ETS factors upon PU.1 induction.

In Fig 5, very minimal changes are detected with respect to accessibility upon interrogation of the PU.1 mutations shown (especially on example tracks).

In fact, the changes in PU.1 binding are marked, and we apologize for not illustrating this better. To make this clearer, we have exchanged Figure 5C for another example region, with better resolution, which shows differences more clearly. To further highlight the magnitude of changes, we rephrased the corresponding paragraph in the main text and included the percentage of peaks with reduced signals, corresponding to the data presented in Fig. 5E.

Finally, while the authors try to identify proximal proteins using BioID and other strategies, they do not validate any of them well or using robust methodology. For example, to evaluate binding to the well-studied SWI/SNF complex (for which they pull out several members), they should have used antibodies toward a number of components, not only BRG1 (in the IP-western blot experiments). They should also validate the other proteins recovered so that the dataset can be evaluated in terms of utility.

We initially focused on BRG1 as the catalytical (and likely most relevant) component of SWI/SNF. In response to this and reviewers #3 comments, we have extended our CoIP analyses to other SWI/SNF components, including ARID2 and SMARCE1 as well as performing the reverse CoIP with BRG1 (now also with the ΔQ mutant). As shown in Figure S6, we detected SWI/SNF components in PU.1-IPs as well as PU.1 in Brg1-IPs. As recommended by the reviewer, we also tested for direct interaction of other PU.1 proximal proteins. However, we could not detect FLI1 or LDB1 in CoIP Westerns, suggesting that, while being proximal to PU.1, they don't interact with PU.1 directly, or that the interaction is not stable during CoIP. These additional experiments extend, confirm and strengthen our previous results, as also discussed further below.

Further, the ATAC-seq and ChIP-seq for BRG1 signals are quite low and the key "points" of the figure are altogether missed.

Again, we apologize for not illustrating this better. The originally shown BTK locus is located on the X chromosome with only half of the read coverage compared to autosomes, which may have left the impression of low signals. We have now replaced the example region and increased resolution of the image to make the “key points” clearer. While we do see residual binding of the ΔA mutant across the 45K PU.1 binding sites, it is unable to recruit SWI/SNF to these sites and fails to induce chromatin accessibility. BRG1 ChIPseq coverage and ATAC signals of this mutant are not different from the control samples (mutPU.1).

Reviewer #2 (Remarks to the Author):

The study of Minderjahn and colleagues investigates how PU.1, a major regulator of hematopoiesis, accesses DNA and re-models adjacent chromatin. They use a combination of analysis of existing and new ChIPseq data to examine PU.1 binding across a swathe of human blood cell lineages and transformed cells, as well as an ectopic expression system. These approaches allow the authors to make some general conclusions about the factors that affect PU.1 DNA binding and transcription regulation. Finally, the authors use Mass Spec approaches to identify SWI/SNF the re-modelling complex that interacts with PU.1 to remodel chromatin at target sites.

Overall this is a thorough and interesting manuscript, that uses some nice approaches to extract useful information out of the global DNA binding data. The data cover some of the same territory as others including Heinz (ref6) and Ghisletti (ref7) and the same authors (ref8, 10), but I feel that the present study clearly extends that earlier work in some important new directions. I feel that this manuscript would be suited for nature Comms with some relatively simple improvements.

1. Concentration of PU.1 in the mRNA transfection model. There is abundant literature to indicate that PU.1 function is concentration dependent, yet the western in figure 2A doesn't show any reference data of endogenous PU.1-expression in other cell lines. I realise that this is not a perfect way of quantifying relative expression, but at least it will give a rough estimate as to whether the expression is low (as in B cells), high (macs) or supraphysiological, as this may impact on some site usage. An alternative quantitation approach could use the relative mRNA levels in RNAseq datasets.

We agree that this information has been missing and now provide a Western Blot of CTV1 transfected with a range of PU.1 mRNA concentrations side-by-side with human cell types expressing PU.1 (monocyte-derived dendritic cells and macrophages, monocytes, as well as THP-1 and K562 cell lines). The expression induced by PU.1 mRNA is certainly in the range of highly expressing cell types and probably exceeds physiological levels at peak times. This is also reflected by the number of peaks detected in transfected CTV-1 cells (45K) and monocyte-derived macrophages (35K), dendritic cells (51K) or PMA & Vitamin D3-treated THP-1 cells (63K). This is now mentioned in the main text.

2. Following on from above, given the impressive number of samples amassed in figure 1A, it would be useful to provide analysis of the numbers of ChIPseq peaks in each cell type and ideally how these correlates (or not) with PU.1 mRNA (or cell type).

We have now added numbers of peaks for all individual experiments in Tables S2 and S3. The suggested correlation of peak numbers and mRNA expression is unfortunately not feasible, since we don't have comparable RNAseq data sets for all cell types. In addition, peak calling depends on read depth and ChIP quality, which vary among published and own data sets. So, while being interesting, the informative value of the suggested correlations would unfortunately be limited.

3. Can the authors clarify what type of cell, CTV-1 cells are representative of. The manuscript is very unclear on this, as is the literature, but as the cells do not express PU.1 it would appear to me that the cells are likely to be of T cell origin.

CTV-1 cells are considered an acute lymphoblastic leukemia. We have now added this information in the main text.

4. Figure 1B. Can the authors clarify in the legend how “no signal” (red) differs from the gray “mappable” area (contains PU.1 consensus site, but no PU.1 bound). This is unclear to me. We made this distinction, because in regions not detected as peaks the coverage of PU.1 reads is quite variable and often just below the detection level in individual cell types. The “no signal” peaks were selected due to their very low read coverage (less than 3 per 10^7 reads within 200-bp motif centered window) across all cells. We have now added this information for clarification.

5. Figure 4F, S5C. Again, I am struggling to understand this graph. Particularly what the % overlap of the mutPU.1 (gray) samples indicates. Could the authors clarify this in the legend? We have revised our legend to make this clear. Grey bars indicate the overlap of ETS/FLI1 peaks in control transfected CTV-1 (in the absence of PU.1) with PU.1 peaks detected in CTV-1 after PU.1 induction. Hence, this graph indicates the massive recruitment of ETS1/FLI1 to PU.1 binding sites, in particular at de novo induced PU.1 binding sites (clusters 1-8).

Reviewer #3 (Remarks to the Author):

This study by Minderjahn and colleagues investigates the property of PU.1, a major transcription factor, in binding and remodeling the chromatin to eventually modify its accessibility. They use a combination of high-throughput genome-wide analyses to investigate the mechanisms by which PU.1 occupies its DNA binding sites and shapes regulatory landscapes. The authors conclude that the PU.1 N-terminal domain binds to the SWI/SNF chromatin remodeling complex and that this interaction acts as a remodeler to control the distribution of transcription factors onto chromatin.

This is an interesting study. The paper is well written and the in vitro analysis of PU.1 access to chromatin is convincing and technically sound. However, although the link between PU.1 and the swi/snf complex is quite appealing and novel, some of the presented data lack of clarity and additional experiments are necessary to allow all the drawing of conclusions as stated in the manuscript. While I understand that it might not be feasible for the authors to address during the time of the revision all of my concerns, attention to the following comments would further improve the manuscript:

Major points:

The author show that PU.1 interacts with the swi/snf complex. They further suggest that (1) this interaction is mediated by the N-terminal AAD domains of PU.1, and (2) this interaction occurs on chromatin.

1) Does PU.1 directly bind to the swi/snf complex? This is suggested by the Flag IP in Fig 6D but should be confirmed by additional experiments.

Do reverse IPs with BRG1 and/or other members of the swi/snf complex pull down PU.1 protein?

Is the PU.1 - swi/snf complex interaction dependent on DNA or is it a direct protein-protein interaction? In case of a direct protein-protein interaction, is it mediated solely by BRG1 – PU.1 binding?

We agree with the reviewer that further validation would strengthen our findings. In response to his/her comment, we performed the reverse CoIP with BRG1 and performed CoIPs with and without benzonase treatment (to digest DNA in cell lysates). As now shown in Figures S6E, we detected BRG1 in PU.1 IPs regardless of the presence of DNA, and as shown in Figures S6F, we were also able to detect PU.1 in reverse BRG1 IPs. While these experiments clearly corroborate our previous finding and suggest that the interaction is protein-protein mediated,

unfortunately, we are unable to say whether the interaction is mediated solely by BRG1, since it is part of the large SWI/SNF complex and other components might mediate or participate in this interaction. However, the precise mapping of the interaction surface would be beyond the scope of the current manuscript.

2) In Fig 6, can authors follow up on the mass spectrometry data by blotting for other members of the swi/snf complex than BRG1 upon Flag-PU.1 immunoprecipitation. PU.1 binding to the swi/snf complex is solely revealed by Pu.1-BRG1 interaction throughout the manuscript.

Why did the author have decided to follow up with BRG1?

Authors claim that the PU.1 - swi/snf complex interaction is not cell line dependent. However; it does not seem to be the case in K562 cells where PU.1 does not bind BRG1 as shown in the volcano plot in Fig S6D. Does PU.1 – BRG1 interaction is significantly enrich the K562 cells.

In line with the current understanding of SWI/SNF as multiprotein complexes, we initially focused on BRG1 as a representative and catalytical (and likely most relevant) component of SWI/SNF complexes. In response to this and reviewers #1 comments, we have extended our CoIP analyses to other SWI/SNF components for which we could identify antibodies that worked in immunoblotting. This included ARID2 and SMARCE1, which were both detected in PU.1 CoIPs (see novel Figures S6G,H). As suggested by reviewer#1, we also tested for direct interaction of other PU.1 proximal proteins. However, we could not detect FLI1 or LDB1 in CoIP Westerns, suggesting that, while being proximal to PU.1, they don't interact with PU.1 directly, or that the interaction is not stable during CoIP.

Authors claim that the PU.1 - swi/snf complex interaction is not cell line dependent. However; it does not seem to be the case in K562 cells where PU.1 does not bind BRG1 as shown in the volcano plot in Fig S6D. Does PU.1 – BRG1 interaction is significantly enrich the K562 cells.

We apologize for the confusion – the K562 BioIDs were “noisier” than those of the other two cell lines. To make this clearer, we have refocused the presentation of MS data on SWI/SNF components and revised the statistical analysis of the corresponding data. Figure S6D now focuses on the enrichment of corresponding peptides in PU.1-BirA mRNA vs. NLS-BirA mRNA transfected cells, indicating that SWI/SNF components are enriched across cell lines. Because the K567 data is more “noisy”, none of the peptides passes the FDR <0.05 after correction. However, the trend for enrichment is obvious.

3) Authors claim that the N-terminal AAD domains of PU.1 interacts with the swi/snf complex. This is mainly supported by the fact that the ΔA PU.1 fragment does not bind to BRG1 or to other members of the swi/snf complex (Fig 6D and S6D). They further claim that ΔA mutant lacks remodeling capacity (Fig 5E).

Assuming that PU.1 interacts with swi/snf on the chromatin, it would be important to access the chromatin binding capability of the ΔA PU.1 fragment. If ΔA binds to chromatin, does it lack the interaction with BRG1 on chromatin?

This is essentially addressed by ChIP experiments shown in Figure 6F,G. The α -Flag ChIPseq coverage across all peaks (Figure 6G, top panel) clearly suggests residual binding of the ΔA PU.1 mutant through clusters 1-14. However, especially at sites that are de novo remodeled by wild-type PU.1 (and also to a large extent by the ΔQ mutant), the coverage for ATAC and α -BRG1 remain at the same level as in control transfected cells. Hence, while being able to bind to its motif in nuclear DNA (at least to some extent), ΔA is not able to recruit SWI/SNF and does not open up the chromatin.

In addition, can authors show the volcano plots in S6D for ΔA PU.1 vs NLS birA.

The volcano plot for the ΔQ mutant is missing as well.

In addition, author should show in Fig 6D that ΔQ mutant behaves like the WT PU.1 protein and bind BRG1.

In response to this comment, we have rearranged the presentation of MS data to focus on SWI/SNF components and revised the statistical analysis of the corresponding data. We feel that the presentation of direct comparisons of wild type, control and mutant constructs is clearer than the set of individual Volcano plots. Figure 6D now specifically focuses on the enrichment of corresponding peptides in PU.1-BirA compared to NLS-BirA, Δ Q-BirA, and Δ A-BirA mRNA transfected cells. In line with the additional ColP data, the Δ Q mutant only shows a slight (non-significant) reduction in SWI/SNF components compared to the wild type PU.1 construct. However, the Δ A mutant shows a strong (and often significant) reduction in SWI/SNF components compared to the wild type PU.1 construct.

4) In the abstract, author say "PU.1's N-terminal acidic activation domain and its ability to recruit SWI/SNF remodeling complexes". However, this statement is not supported by experiments. Authors should assess the swi/snf complex recruitment to the chromatin upon inhibition of PU.1. The reciprocal experiments shown in S6E-F does not directly support their claim.

This reviewer suggests that we should inhibit PU.1 binding to assess SWI/SNF complex recruitment to chromatin to prove the above statement. However, we feel that the recommended approach will not substantiate our claim further for two main reasons: Firstly, it is not necessary to knock-down or inhibit PU.1 in a cell line that does not express PU.1. In our case the mutPU.1 transfected CTV-1 cell line (which does not express PU.1) serves as the suggested control experiment and BRG1 is only recruited to *de novo*-remodeled sites upon PU.1 induction.

Secondly, while it would be possible to inhibit PU.1 binding in a cell line that constitutively expresses PU.1 (like e.g. THP-1 cells), this type of experiment would address different questions. In contrast to our setting, which studies *de novo* binding and remodeling, the suggested experiment would inform about sites which require PU.1 binding to maintain accessible chromatin (provided that the cell survives PU.1 depletion). The mechanisms behind *a*) opening up sites *de novo* and *b*) maintaining sites in open chromatin are likely different.

While it is clear that experiments inhibiting BRG1 (as now shown in Fig. S6I,J) are not directly supporting our claim (because we see a general effect on chromatin accessibility), we feel that the large body of evidence (including the novel data) provided in this study now sufficiently supports our claims.

5) Regarding the BirA experiments, can the author provide with westernblot for global biotination with and without the 50 μ M biotin treatment for all cell lines used for mass spectrometry.

As requested, we now include Western blots for global biotinylation with and without the 50 μ M biotin treatment for the CTV-1 cell line transfected with the various constructs. The data has been added as Figure S6A. As expected, the staining increases with the addition of biotin.

Minor points:

1) PU.1 Ab is required in Fig 6D.

As requested, for ColP Western blots shown in Figure 6 and S6, we now included anti-PU.1 staining in addition to anti-FLAG staining.

2) In Fig 6B, can the author explain if log₂ fold change refers to iBAQ or intensity?

The log₂ fold change refers to iBAQ values, which is now mentioned in the corresponding figure legend.

3) In Fig 6B, SMARCA2 labeling is misleading. The protein seems depleted when it is enrich in PU.1 immunoprecipitated samples

To make this clearer, we have increased the line width connecting the labels with data points. It should now be clear that SMARCA2 is enriched (and not depleted)

REVIEWERS' COMMENTS:

Reviewer #2 (Remarks to the Author):

I feel that the authors have adequately addressed my comments.

Stephen Nutt

Reviewer #3 (Remarks to the Author):

In the new version of the manuscript, the authors have addressed (partially) all my concerns. Authors provide a set of new data that, according to their conclusions, confirm and strengthen their previous findings. In my opinion, the manuscript is suitable for publication.

Prior to publication, authors should make efforts to clarify this last point:

In Fig 6SE, Flag IP is done in presence or absence of benzonase. However, H3 is not observed in the pulldown done without benzonase as it is seen in the Flag IP shown in 6D. This is concerning.

Response to the final comments of reviewers:

Point-by-point response:

Reviewer #3 (Remarks to the Author):

In Fig 6SE, Flag IP is done in presence or absence of benzonase. However, H3 is not observed in the pulldown done without benzonase at it is seen in the Flag IP shown in 6D. This is concerning.

The original figure was chosen because we didn't want to overexpose the input signal and we can still see the bands (in the high resolution image), but we agree, the signal intensity may be weak. To resolve this issue, this image has now been replaced with an image from the same blot with a longer exposure time where the H3 bands are visible more clearly.